# Proteasomal degradation of NOD2 by NLRP12 in monocytes promotes bacterial tolerance and colonization by enteropathogens

Sylvain Normand[1], Nadine Waldschmitt[1,2], Andreas Neerincx[3], Ruben Julio Martinez-Torres[4], Camille Chauvin[1], Aurélie Couturier-Maillard[1], Olivier Boulard[1], Laetitia Cobret[5,6], Fawaz Awad[5,6], Ludovic Huot[1], Andre Ribeiro-Ribeiro[4], Katja Lautz[7], Richard Ruez[1], Myriam Delacre[1], Clovis Bondu[1], Martin Guilliams [8,9], Charlotte Scott[8,9], Anthony Segal[4], Serge Amselem[5,6], David Hot [1], Sonia Karabina[5,6], Erwin Bohn[10], Bernhard Ryffel[11], Lionel F. Poulin[1], Thomas A. Kufer [12] & Mathias Chamaillard[1]

Mutations in the nucleotide-binding oligomerization domain protein 12 (NLRP12) cause recurrent episodes of serosal inflammation. Here we show that NLRP12 efficiently sequesters HSP90 and promotes K48-linked ubiquitination and degradation of NOD2 in response to bacterial muramyl dipeptide (MDP). This interaction is mediated by the linker-region proximal to the nucleotide-binding domain of NLRP12. Consequently, the disease-causing NLRP12 R284X mutation fails to repress MDP-induced NF-κB and subsequent activity of the JAK/STAT signaling pathway. While NLRP12 deficiency renders septic mice highly susceptible towards MDP, a sustained sensing of MDP through NOD2 is observed among monocytes lacking NLRP12. This loss of tolerance in monocytes results in greater colonization resistance towards *Citrobacter rodentium*. Our data show that this is a consequence of NOD2-dependent accumulation of inflammatory mononuclear cells that correlates with induction of interferon-stimulated genes. Our study unveils a relevant process of tolerance towards the gut microbiota that is exploited by an attaching/effacing enteric pathogen.

[1] University of Lille, CNRS, Inserm, CHU Lille, Institut Pasteur de Lille, U1019 - UMR 8204 - CIIL - Centre d'Infection et d'Immunité de Lille, F-59000 Lille, France. [2] Technische Universität München, Chair of Nutrition and Immunology, 85350 Freising-Weihenstephan, Germany. [3] Department of Pathology, University of Cambridge, Tennis Court Road, Cambridge CB2 1QP, UK. [4] Division of Medicine, University College London, WC1E 6BT London, UK. [5] Sorbonne Universités, UPMC Univ Paris 06, UMR_S 933, F-75012 Paris, France. [6] Inserm, UMR_S 933, F-75012 Paris, France. [7] Institute for Medical Microbiology, Immunology and Hygiene, University of Cologne, Cologne, Germany. [8] Laboratory of Immunoregulation, VIB Inflammation Research Center, 9052 Ghent, Belgium. [9] Department of Internal Medicine, Ghent University, Ghent 9000, Belgium. [10] Interfakultaeres Institut für Mikrobiologie und Infektionsmedizin, Eberhard Karl Universitat Tuebingen, 72076 Tuebingen, Germany. [11] CNRS, Orléans University, INEM, UMR 7355, F-45071 Orléans, France. [12] Department of Immunology, Institute of Nutritional Medicine, University of Hohenheim, Stuttgart, Germany. These authors contributed equally: Sylvain Normand, Nadine Waldschmitt, Andreas Neerincx, Ruben Julio Martinez-Torres. These authors jointly supervised this work: Lionel F. Poulin, Thomas A. Kufer, Mathias Chamaillard. Correspondence and requests for materials should be addressed to M.C. (email: mathias.chamaillard@inserm.fr)

Mutations of the nucleotide-binding oligomerization domain protein 12 (NLRP12) are causing a familial cold-induced auto-inflammatory syndrome (referred as FCAS2; OMIM 611762) that belongs to the group of hereditary recurrent fevers[1]. The aforementioned Mendelian disorders are primarily characterized by intermittent episodes of fever and serosal inflammation (including sterile peritonitis, arthritis, and abdominal pains) that may coincide with myalgia and urticarial rash. In contrast to most members of the nucleotide-binding domain leucine-rich repeat proteins (NLR) family, NLRP12 (also known as NALP12, RNO, MONARCH-1, and PYPAF-7) is thought to play a suppressive role on inflammatory responses. Indeed, overexpression of FCAS2-causing *NLRP12* frameshift mutations results in unrestrained activation of nuclear factor kappa-light-chain-enhancer of activated B cells (NF-κB)[1]. Besides this inhibitory role on the canonical NF-κB pathway, biochemical studies revealed that NLRP12 also negatively interferes with the proteasome-mediated degradation of NF-κB-inducing kinase (NIK) through the TNF receptor associated factor 3 (TRAF3) in human monocytes[2,3]. Foremost, genetic ablation of *Nlrp12* renders mice highly susceptible to colitis and colitis-associated colorectal cancer[4,5], while being resistant to Salmonellosis[6]. However, the function of NLRP12 on bacterial tolerance and host defense remains largely unappreciated as mutant animals were also found highly susceptible to infection by a vaccinated strain of *Yersinia pestis*[7]. Collectively, these paradigms argue for the need to better understand the complex regulatory mechanisms by which NLRP12 signaling could mediate tolerance to the gut microbiota while being exploited by some bacterial pathogens[7].

*Escherichia coli* is a versatile Gram-negative commensal that typically colonizes the gastrointestinal tract within a few hours after birth. *E. coli* successfully exploits several ways to efficiently compete with other microorganisms allowing it to occupy specific niches in the gut. Enterohemorrhagic *E. coli* (EHEC) and enteropathogenic *E. coli* (EPEC) remains an important cause of diarrhea or hemorrhagic colitis in humans worldwide[8]. Infection by EPEC and EHEC results in the effacement of the brush border microvilli followed by bacterial attachment to the apical plasma membrane of intestinal epithelial cells. Similar to the related EPEC and EHEC, *Citrobacter rodentium* is an extracellular enteric bacterial pathogen that naturally colonizes the caecum and the colon of mice[9]. As is observed in humans, *C. rodentium* attaches to and colonizes the intestinal epithelium by triggering the development of lesions and infiltration of phagocytic mononuclear cells[10]. Of note, protective immunity to *C. rodentium* involves several Crohn's disease predisposing genes, among which are the nucleotide-binding oligomerization domain containing protein 2 (encoded by the *NOD2* gene)[11] and the autophagy 16-like 1 (encoded by the *ATG16L1* gene). NOD2 is required for local production of the chemokine CCL2 through the recruitment of the serine-threonine kinase RIPK2 (also known as CARDIAK and RIP2)[12]. Indeed, a prolonged bacterial shedding in infected *Nod2*-deficient mice results at least partially from an impaired recruitment of monocytes[11]. It is also worth noting that ATG16L1 was found to interact with NOD2[13] and to interfere with poly-ubiquitination of the serine-threonine kinase RIPK2[14] that is required for NF-κB activation in response to bacterial muramyl dipeptide (MDP). Consequently, animals that are hypomorphic for ATG16L1 expression showed enhanced NOD2-mediated protection against *C. rodentium*[15].

Herein, we report evidence that NLRP12 promotes bacterial tolerance via degradation of NOD2 through the Ubiquitin–Proteasome Pathway. This involves sequestration of the heat-shock protein 90 (HSP90) that is one of the most abundant molecular chaperone in the cytosol. Loss of HSP90 recruitment results in MDP tolerance through a failure to protect NOD2 from being degraded by the proteasome[16]. This loss of MDP tolerance results in enhanced NOD2-dependent recruitment of inflammatory monocytes that is associated with the induction of several interferon-stimulated genes (ISG), including IFIT2 that was found to protect from an attaching-and-effacing bacterial pathogen.

## Results

### The nucleotide binding structures of NLRP12 bind to NOD2.
Since NLRP12 may potentially interact with both Caspase-activating recruitment domains (CARDs) of NOD2 but not that of NOD1[17], we hypothesized that NLRP12 may promote MDP tolerance by dissociating the NOD2-HSP90 complex, which is required for NF-κB activation in response to bacteria[16]. To this end, we generated THP-1 cells stably expressing the fusion protein Myc-BirA*-NOD2. In accordance with yeast two-hybrid screen data[17], we confirmed the NOD2-NLRP12 interaction in monocytic cell line when specifically inhibiting the proteasome degradation of NOD2 that is induced in response to MDP (Fig. 1a). By contrast, the formation of the NOD2–NLRP12 complex was not observed in response to bacterial lipopolysaccharide (LPS) that is neither sensed by NOD2 nor NLRP12 (Fig. 1a and Supplementary Fig. 1). HEK293T cells were next transfected with plasmids transiently expressing FLAG-tagged NOD2 and Myc-tagged NLRP12 (Fig. 1b). Overexpression of full-length FLAG-tagged NOD2 together with Myc-tagged NLRP12 resulted in an interaction with RIPK2, which was underrepresented in the complex at high NLRP12 concentration as shown by immunoprecipitation using an anti-FLAG antibody (Fig. 1c). Mapping of the interaction domain from human NLRP12 further showed that the assembly of a protein complex with NOD2 involved a linker region of NLRP12 (residues 200–224) in which ATP binding is required for the inhibition to occur (Fig. 1d)[2]. By contrast, the N-terminal Pyrin domain of NLRP12 (residues 1–98) that is required for apoptotic signaling[18] was found dispensable for specifically interacting with NOD2 (Fig. 1d and Supplementary Fig. 2). Together, these results suggested that NLRP12 might interfere with NOD2 signaling in monocyte-derived cells where NLRP12 is primarily expressed (Supplementary Fig. 3).

### NLRP12 triggers MDP tolerance through degradation of NOD2.
To corroborate the role of NLRP12 as a potential checkpoint blocker of NOD2 signaling in monocytes, we examined the influence of NLPR12 on the stability and the activity of the NOD2/RIPK2 complex. Co-immunoprecipitation experiments revealed that NLRP12 expression promotes poly-ubiquitination of the NOD2/RIPK2 complex in HEK293T cells (Fig. 2a and Supplementary Fig. 4). By contrast, the ubiquitination status of NOD1 was not influenced by full-length NLRP12 (Supplementary Fig. 5a). Consistent with a regulation of NOD2 activity at the protein level through its interaction with NLRP12, blocking of protein neosynthesis using cycloheximide revealed that NLRP12 expression reduced the half-live of the NOD2 protein (Fig. 2b). This inhibitory effect of NLRP12 on the stability of NOD2 was compromised upon inhibition of the ubiquitin-proteasome pathway by MG132 (Supplementary Fig. 5b). Accordingly, transient expression of full-length human NLRP12 greatly reduced NOD2-mediated p50/p65 reporter activation in response to MDP in HEK293T cells (Fig. 2c). While increasing amounts of transfected NLRP12 further reduced MDP-induced NF-κB activation by about 70% (Fig. 2c), this correlated with the lowered protein levels of both NOD2 and RIPK2 when NLRP12 is overexpressed (Fig. 1c). The formation of such protein complexes between NOD2 and NLRP12 coincided with a greater assembling of K48 poly-ubiquitin chains on NOD2 when pretreating THP-1 cells stably expressing the fusion protein Myc-BirA*-NOD2 with the proteasome inhibitor MG132 (Fig. 2d and Supplementary

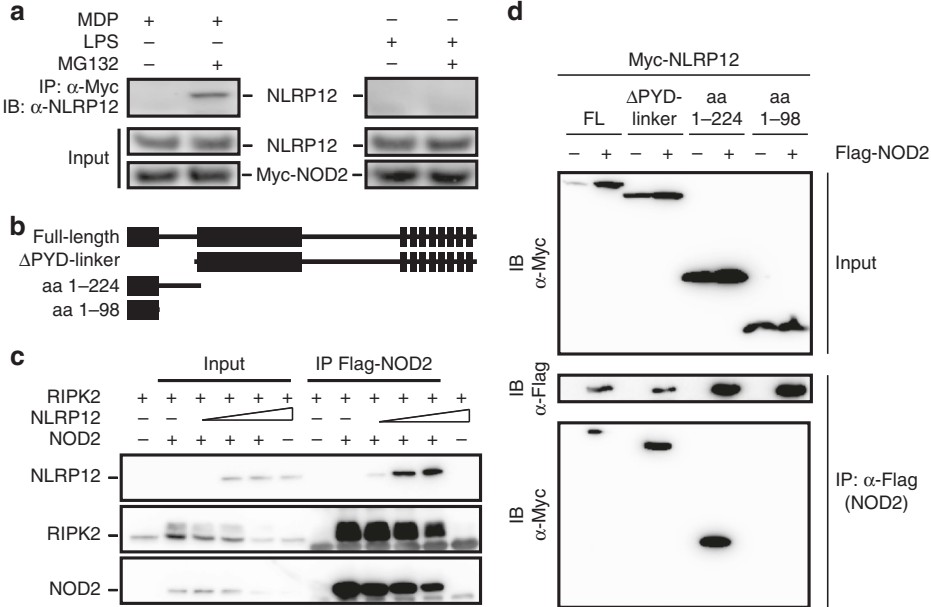

**Fig. 1** NLRP12 interacts with NOD2 through a linker-region proximal to the nucleotide-binding domain. **a** Co-immunoprecipitation with anti-Myc (top panel) and lysates (bottom panel) of the THP-1 Myc-BirA*-NOD2 stable cell line that were treated with muramyl dipeptide (MDP at 10 μg/mL) or lipopolysaccharide (LPS at 0.2 μg/mL) for two h. **b** Scheme of Myc-tagged constructs encoding human wild-type NLRP12 and mutated NLRP12 proteins Δ-PYD linker (200 aa–1061 aa), 1–224 aa and 1–98 aa. **c** Western blot analysis of NOD2–RIPK2 complexes that were precipitated in the presence of increasing amounts of NLRP12 in HEK293T cells. **d** Representative co-immunoprecipitation experiment with indicated plasmids. Immunoprecipitation was performed on native protein lysates by using antibodies against FLAG (M2). NOD2-specific immunoprecipitation was confirmed by western blot analysis using anti-Flag antibody. NLRP12 and NOD2 were detected using anti-Myc 9E10 and anti-FLAG (M2) antibody respectively. Non-precipitated protein lysates were used as input controls

Fig. 6 and 7). As reported previously[19], co-immunoprecipitation experiments confirmed an interaction between NOD2 and the chaperone protein HSP90 in response to MDP that was critical for the inhibition to occur (Fig. 2d). Indeed, pretreating THP1-cells with MG132 strongly inhibited the formation of the NOD2/HSP90 complex in response to MDP, while LPS stimulation expectedly failed to do so (Fig. 2d). Consequently, NOD2 was migrating at higher molecular weights (visible as a smear) when treating THP-1 cells with MDP (Supplementary Fig. 8a). This correlated with a greater amount of NLRP12 as determined by western blotting (Supplementary Fig. 8b). Collectively, we identified NLRP12 as a NOD2-interacting protein that may promote degradation of NOD2 through the Ubiquitin–Proteasome Pathway and subsequently MDP tolerance in monocytes.

**NLRP12 deficiency impairs MDP tolerance in mice**. A lowered *NLRP12* gene expression is observed in patients with septic shock[20], suggesting a potential feedback regulatory loop on NLRP12 function during sepsis. Indeed, endotoxin treatment negatively regulates *NLRP12* promoter activity in human monocytes through the PR domain zinc finger protein 1 (PRDM1)[21]. Given that knocking-down NLRP12 expression may enhance TLR4 signaling in vitro[3], we next aimed to confirm whether *Nlrp12* expression might indirectly influence host responsiveness to bacterial LPS in vivo. To this end, we made use of *Nlrp12*-deficient mice that were generated by replacing the second exon encoding the Pyrin domain of NLRP12 with a neomycin selection cassette through homologous recombination (Supplementary Fig. 9). Wild-type and mutant mice were challenged with a lethal dose of highly purified LPS from *Escherichia coli* 0111:B4, as a model of acute endotoxin septic shock. Unlike *Caspase11*- and *Tlr4*-deficient mice[22], no difference in survival of wild-type and *Nlrp12*-deficient mice was observed (Fig. 3a). We next tested the potential role of NLRP12 in the susceptibility of LPS-sensitized

mice to a secondary challenge with MDP[23]. Wild-type and *Nlrp12*-deficient mice were primed with a non-lethal dose of highly purified LPS from *E coli* 0111:B4. As a consequence of a failure to negatively regulate NOD2 signaling, LPS-primed *Nlrp12*-deficient mice were significantly more susceptible to secondary MDP challenge when compared to similarly treated control animals (Fig. 3b). We next generated mice that are deficient for both *Nod2* and *Nlrp12* and compared the survival rate of this compound mutant mice to the one of *Nlrp12*-deficient mice. Consistent with our in vitro findings, a greater survival rate was observed in animals lacking both NOD2 and NLRP12 (Fig. 3c). Thus, NLRP12 is dispensable for protecting mice against endotoxemia, but rather function as a negative regulator of NOD2 signaling in mice.

**MDP tolerance is lost in monocytes deficient for NLRP12**. Given that unrestrained human NOD2 signaling results in autoinflammatory syndromes[24], we next assessed whether the FCAS2-causing mutation in the *NLRP12* gene may fail to dominantly repress activation of NF-κB in response to MDP. To this end, we generated plasmids encoding the most frequent mutations in *NLRP12*. Each mutant was transiently transfected in HEK293T cells together with full-length NOD2. All mutants with single amino-acid replacements were found to efficiently repress MDP-induced activation of NF-κB (Fig. 4a) and to interact with HSP90 (Fig. 4b), as observed in cells expressing wild-type NLRP12. In contrast, the R284X nonsense mutation failed to inhibit the activation of NF-κB in response to MDP (Fig. 4a) and of the JAK/STAT pathway by the S/T kinase TANK-binding kinase 1 that is commonly referred as TBK-1 (Supplementary Fig. 10). It coincided with a barely detectable recruitment of HSP90 by such mutation (Fig. 4c) even if this was not related to a failure of the R284X nonsense mutation to interact with NOD2 (Fig. 4d). As a consequence, loss of *NLRP12* expression by

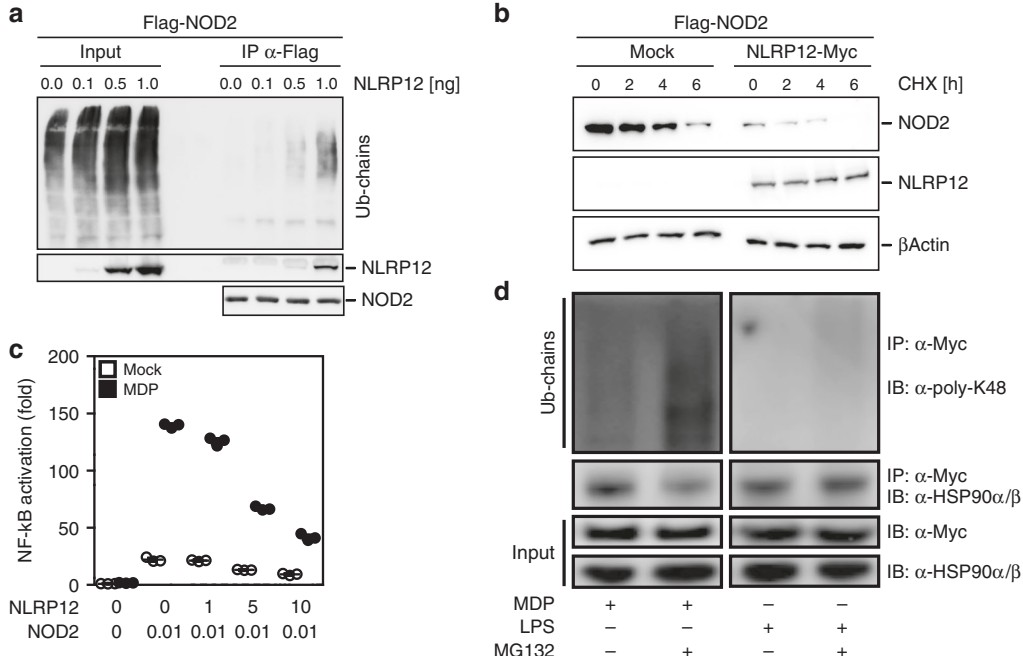

**Fig. 2** NLRP12 suppresses MDP-induced NF-κB activation by targeting the NOD2/RIPK2 complex for assembling of K48 poly-ubiquitin chains on NOD2 and subsequent degradation. **a** Western blot analysis of NOD2 ubiquitination in HEK293T cells that were transfected with FLAG-tagged NOD2 in the presence of increasing amounts of NLRP12 as indicated. **b** Western blot analysis of NOD2 stability in the presence of cycloheximide (CHX at 20 μg/mL) for 4 h. HEK293T cell extracts were subjected to western blot analysis for FLAG-tagged NOD2, Myc-tagged NLRP12 and β-actin. **c** NF-κB-Luciferase activity in HEK293T cells normalized to expression of β-galactosidase. Depicted are mean ± SEM ($n = 3$). **d** Immunoblot analysis of poly-K48 ubiquitination on NOD2 and of HSP90α/β following co-immunoprecipitation with NOD2 by using the anti-Myc antibody in THP-1 Myc-BirA*-NOD2 cells that were stimulated with MDP (10 μg/mL; left panel) or LPS (0.2 μg/mL, right panel) for 2 h with or without MG132 (12.5 μM)

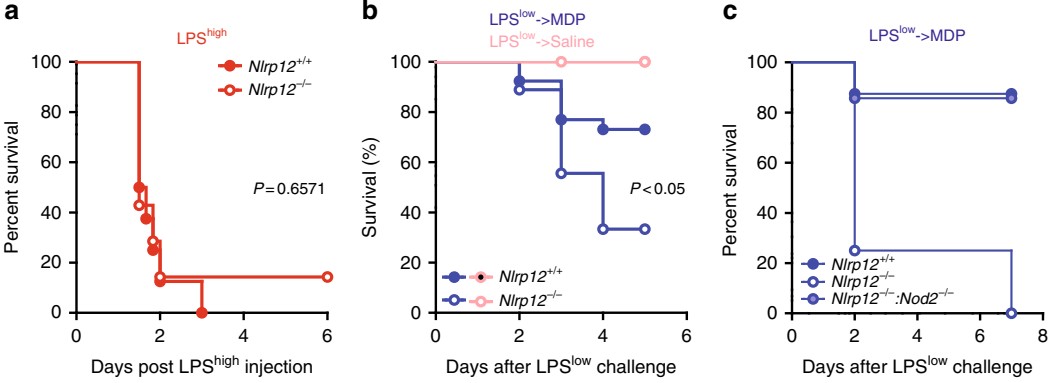

**Fig. 3** LPS-primed *Nlrp12*-deficient mice are highly susceptible to secondary challenge by bacterial MDP. **a** The survival of wild-type mice ($n = 8$) and *Nlrp12*$^{-/-}$ mice ($n = 7$) was plotted after i.p. administration of a lethal dose of ultrapure LPS from *E. coli* O111:B4 (54 mg/kg). **b** The survival of wild-type mice ($n = 18$) and *Nlrp12*-deficient mice ($n = 9$) was plotted after a non-lethal dose of ultrapure LPS from *E. coli* O111:B4 (10 mg/kg) that was followed 24 h later with i.p. Murabutide administration at 10 mg/kg. **c** The survival of wild-type mice ($n = 8$), *Nlrp12*$^{-/-}$ mice ($n = 7$) and compound mutant mice ($n = 8$) was plotted after a non-lethal dose of ultrapure LPS from *E. coli* O111:B4 (2 mg/kg) that was followed 24 h later with i.p. Murabutide administration at 10 mg/kg. *P*-value by log-rank test

CRISPR/Cas9-mediated knockout system in human monocytic THP-1 cells enhanced secretion of TNF-α in response to MDP when compared to parental cells (Fig. 4e). Likewise, MDP induced a greater secretion of both tumor necrosis factor alpha (TNF-α) and interleukin-6 (IL-6) in PBMCs from patients bearing the R284X nonsense mutation when compared to control cells (Fig. 4f and Supplementary Fig. 11a-b). This provided a potential explanation for the loss of tolerance to MDP that account for the failure to detect the truncated protein in such mutant cells by western blotting (Supplementary Fig. 11c). The possibility of a dominant negative effect was further ruled out by

confirming that the protein level of the full-length isoform of NLRP12 was lowered by about 30% in PBMCs from the twin patients bearing the R284X nonsense mutation when compared to control lysates (Supplementary Fig. 11c). Thus, such lack of MDP tolerance was the consequence of haploinsufficiency resulting from the activation of a surveillance pathway referred to as nonsense-mediated mRNA decay, which was blocked by cycloheximide (Fig. 4g). Meanwhile, a greater activation of the NF-κB transcriptional complex by MDP was observed in monocytes from the bone marrow of *Nlrp12*-deficient mice but not in those from *Nlrp12*$^{-/-}$:*Nod2*$^{-/-}$ mice (Supplementary

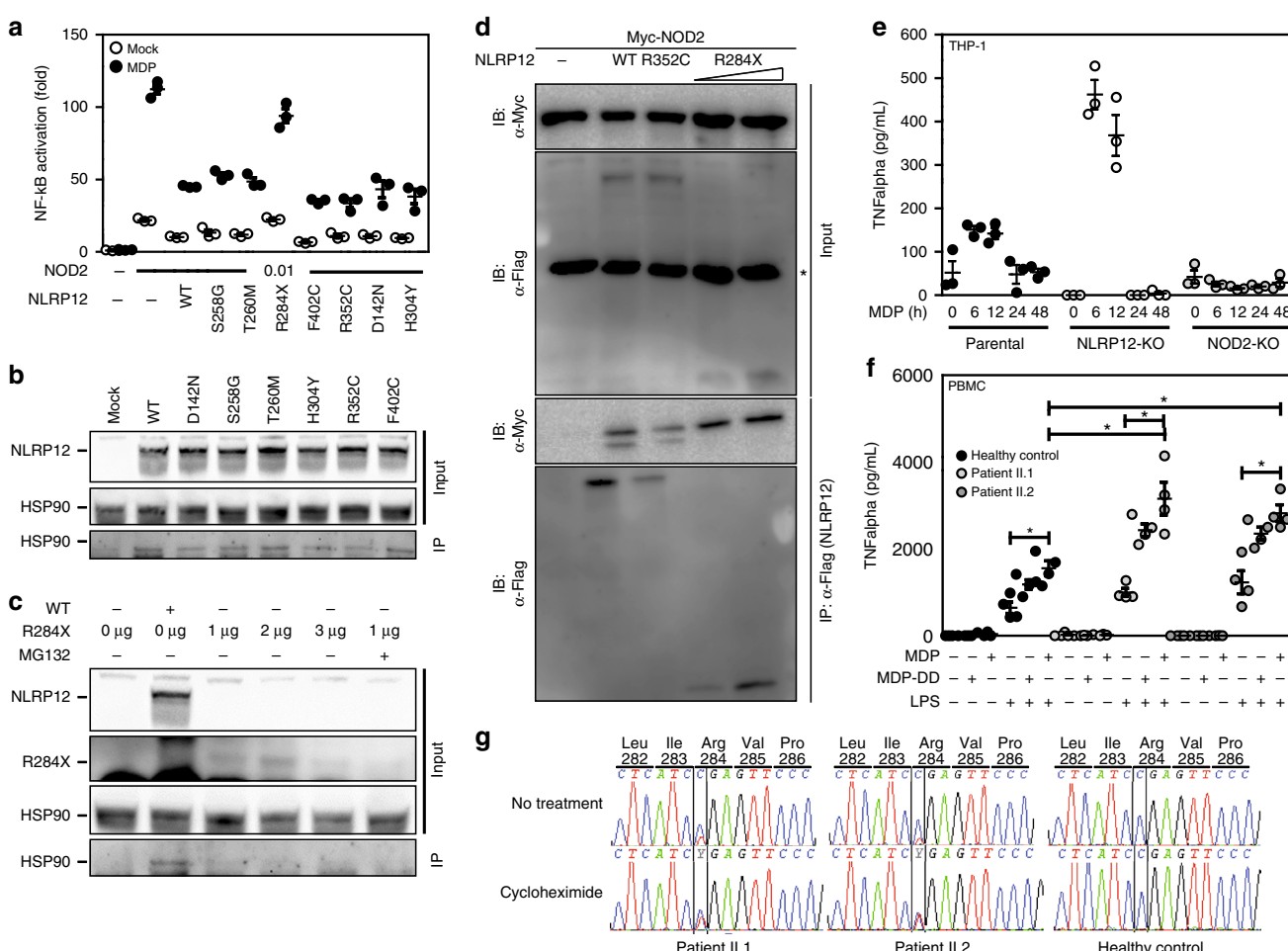

**Fig. 4** The FCAS2-causing mutation results in NLRP12 haploinsufficiency that impairs MDP tolerance. **a** The FCAS2-causing NLRP12 mutant shows impaired NOD2 suppression activity. HEK293T-based NF-κB-Luciferase Assay as described in Fig. 2c. Depicted are mean ± SEM (*n* = 3). **b** NLRP12 interacts with the chaperone protein HSP90. HEK293T cells were transfected with Myc-tagged NLRP12 constructs for 48 h. Myc-tagged NLRP12 was precipitated from lysates and investigated for interaction with HSP90 using western blot analysis. **c** The R284X mutant failed to interact with the chaperone protein HSP90. HEK293T cells were transfected with Myc-tagged NLRP12 constructs for 48 h. Myc-tagged NLRP12 was precipitated from lysates and investigated for interaction with HSP90 using western blot analysis. **d** Representative co-immunoprecipitation experiment with the indicated plasmids. NLRP12-specific immunoprecipitation was performed on native protein lysates by western blot analysis using anti-FLAG (M2) antibody. Immunoprecipitation by using antibodies against FLAG (M2). NLRP12 and RIPK2 complexes with NOD2 were detected by immunoblot (IB) using anti-NLRP12 antibody. Non-precipitated protein lysates were used as input controls. The symbol * refers to non-specific band. **e** MDP-induced secretion of TNF-alpha by THP-1 parental (wild-type), THP-1 NLRP12$^{-/-}$ and THP-1 NOD2$^{-/-}$ cell lines by ELISA. The cells were incubated for either 6 h, 12 h, 24 h or 48 h with MDP at 10 μg/mL before being spun down for collecting the supernatant. All experiments were performed in triplicate. **f** ELISA analysis of TNF-alpha secretion by PBMCs of healthy donors and the twin patients carrying the nonsense R284X mutation in the *NLRP12* gene. **g** Sequencing electropherograms of *NLRP12* cDNA are depicted before and after treatment of patient's PBMCs by 30 μg/mL of cycloheximide (CHX) for blocking nonsense-mediated mRNA decay

Fig. 12a). Consequently, a greater secretion of both TNF-α and IL-6 was also detectable in the supernatant of LPS-primed monocytes that were subsequently treated by MDP (Supplementary Fig. 12b). Likewise, similar findings were observed in macrophages that were derived from the bone marrow of *Nlrp12*-deficient mice (Supplementary Fig. 12c), even if the difference was less pronounced when compared to what observed in monocytes (Supplementary Fig. 12a). A potential explanation for this difference may result from the down-regulation of NLRP12 during differentiation of monocytes into macrophages[21]. Collectively, these results suggested that the serosal inflammation of FCAS2 patients might result from an impaired tolerance of monocytes towards MDP.

**NOD2-dependent ISG response in the gut of *Nlrp12*$^{-/-}$ mice.**
We next examined at high-resolution the colon and caecum of

*Nlrp12*-deficient mice for any spontaneous signs of inflammation as a consequence of an impaired tolerance to MDP derived from the gut microbiota. To this end, whole-genome expression analysis was performed on RNA extracted from the caecum of wild-type and mutant mice. A set of 106 genes were differentially regulated as a result of NLRP12 deficiency, among which 35 genes were significantly up-regulated by at least one log2-fold change in *Nlrp12*-deficient mice compared to controls (Supplementary Data 1). Of these, gene-ontology analysis identified a significant overrepresentation of ISG (Fig. 5a), including those encoding interferon-induced protein 44 (IFI44), interferon-induced protein with tetratricopeptide repeats 2 (IFIT2), Apolipoprotein L9 (APOL9a/b) and 2'-5'-Oligoadenylate synthetase 2 (OAS2). Transcriptional activation of the promoters of those ISG is allowed through the binding to Interferon Stimulated Response Elements (ISRE) of the heteromeric IFN-stimulated gene factor 3

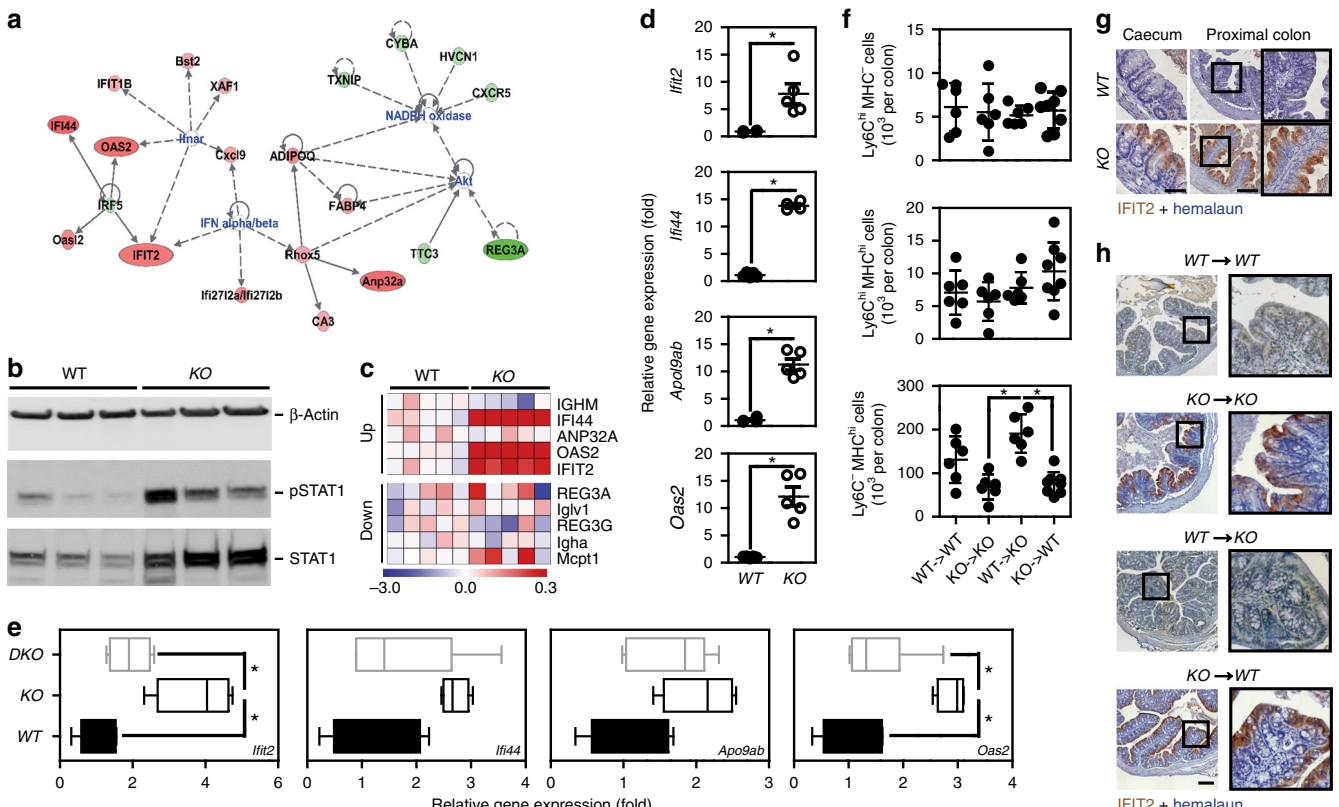

**Fig. 5** Loss of NLRP12 in leukocytes triggers ISG induction within the intestinal epithelium. **a** Network analysis of differentially expressed genes (with log2 fc>1.5) in the caecum of *Nlrp12−/−* mice. **b** Detection of STAT1 expression and activation by western blotting using total tissue samples from caecum of *Nlrp12−/−* mice and wild-type controls (*n* = 3). β-Actin was used for loading control. **c** Top 5 up- and downregulated genes in the caecum of *Nlrp12−/−* mice (*p* < 0.01) when compared to controls. **d** Validation of microarray-based gene regulation by RT-qPCR analysis (*n* = 5). **e** Relative gene expression of *Ifi44, Ifit2, Oas2,* and *Apol9a/b* was analyzed in the proximal colon of wild-type and mutant mice. Statistical significance was calculated by non-parametric Mann–Whitney test. *P* < 0.05 (*) was considered statistically significant. **f** Immunohistochemistry for IFIT2 was performed on 5 μm-thick tissue sections from caecum, proximal and distal colon of naïve wild-type and *Nlrp12−/−* mice. Scale bars represent 50 μm (caecum) and 200 μm (proximal colon). **g** IFIT2 protein expression was analyzed by IHC on tissue sections prepared from proximal colon of wild-type chimeric mice (WT→WT), *Nlrp12−/−* recipients that were reconstituted with hematopoietic cells from mutant mice (KO→KO), wild-type recipients that were reconstituted with hematopoietic cells from *Nlrp12−/−* mice (KO→ WT), and *Nlrp12−/−* mice that were reconstituted with hematopoietic cells from wild-type mice (WT→KO). Scale bars represent 200 μm. **h** Cytofluorometry analysis of living Ly6C^low MHCII^high (commonly referred as macrophages) that were isolated from the colon of chimeric mice (*n* = 6–8) following exclusion of NK cells, B cells, T cells, eosinophils, and neutrophils as previously described. Statistical significance was assessed by non-parametric Mann–Whitney test. *P* < 0.05 (*) and *P* < 0.01 (**) was considered statistically significant

complex (ISGF3) that is composed of the signal transducer and activator of transcription 1 (STAT1). In agreement with this, a tonic activation of the transcription factor STAT1 was observed in the caecum of *Nlrp12*-deficient mice when compared to that in controls (Fig. 5b), while it was lowered in the absence of NOD2 signaling at steady state (Supplementary Fig. 13a). Consistently, qRT-PCR analysis confirmed a greater expression of the aforementioned differentially expressed genes (namely *Ifi44, Ifit2, Oas2,* and *Apol9a/b*) in the caecum from *Nlrp12*-deficient mice (Fig. 5c, d), but not in the intestine of either *Nlrp12−/−:Nod2−/−* or *Nod2−/−* mice (Fig. 5e and Supplementary Fig. 13b respectively). Further, the use of the ISRE-luciferase reporter gene assay led us to reveal that overexpression of NLRP12 significantly lowered the activation of the JAK/STAT signaling pathway by TBK-1 (Supplementary Fig. 10). To further understand the sequence of events leading to this robust antiviral response at baseline, RNA was next extracted from isolated intestinal epithelial cells (IECs) and from *lamina propria* mononuclear cells (LPMCs) of wild-type and mutant intestine. In line with previous findings, the transcript level of the above-mentioned differentially expressed genes (including *Ifi44, Ifit2, Oas2,* and *Apol9a/b*) was

primarily enriched in primary IECs isolated from both the colon and the caecum of *Nlrp12*-deficient mice (Supplementary Fig. 14a). Consistently, immunohistochemical analysis revealed an enhanced production of either IFIT2 or OAS2 that is essentially restricted to the epithelium of the caecum and proximal colon from *Nlrp12*-deficient mice (Fig. 5f and Supplementary Fig. 15). This is in line with the idea that NOD2-mediated inflammasome activation is enhanced by IL-32[25], which subsequently triggers type I/III interferon (IFN) activation[26,27]. In contrast, the transcript levels of numerous genes with ascribed function on mucosal adaptive immunity (including *Ighm, Ptprc, Anp32a,* and *Mcpt1*) were not significantly changed (Supplementary Fig. 14b). To specifically elucidate the cell type responsible for the negative effect of NLRP12 on the epithelial expression of IFIT2[28], we generated bone-marrow chimeras. While the expression of IFIT2 was barely detectable in the proximal colon of wild-type chimeric mice (WT→WT) (Fig. 5g), the intestinal epithelium of *Nlrp12*-deficient recipients that were reconstituted with hematopoietic cells from mutant mice (KO→KO) was characterized by higher protein level of IFIT2 (Fig. 5g) as what observed in non-chimeric mutant mice (Fig. 5f).

Surprisingly, the epithelial expression of IFIT2 was observed in wild-type recipients that were adoptively transplanted with leukocytes from *Nlrp12*-deficient mice (*KO→ WT*) (Fig. 5g). This coincided with a lowered abundance of macrophages in the intestine 8 weeks after reconstitution of wild-type and mutant recipients with *Nlrp12*-deficient leukocytes, compared to animals that received wild-type bone marrow cells (Fig. 5h and Supplementary Fig. 16). By contrast, transfer of wild-type bone marrow cells into mutant mice (*WT→KO*) failed to induce epithelial expression of IFIT2 (Fig. 5g). Similar results were observed in the caecum of chimeric mice (Supplementary Fig. 16). In accordance with our data in the model of acute endotoxin septic shock (Fig. 3), IFIT2 primarily functions as a downstream effector of the IFN-λ receptor[28] that may subsequently influence the outcome of *Nlrp12*-deficient mice in response to endotoxin shock[29]. While IFIT2 is thought to regulate cell death and inflammation[29], epithelial proliferation is also orchestrated by several additional ISG that were upregulated within the epithelium of *Nlrp12*-deficient mice[30], such as Apol9a/b[31]. In this context, it is worth noting that the effect of IFN alpha/beta on IFIT2 induction is far less compartmentalized than the one of the IFN lambda receptor that is primarily expressed to epithelial cells[32]. Altogether, the bone marrow experiments demonstrated that NLRP12 deficiency in leukocytes is responsible for a greater inflammatory response driven by type I/III interferons downstream of NOD2 signaling in monocytes. This suggested us such epithelial ISG induction is likely the consequence of greater NOD2-dependent secretion of type I/III interferons by monocytes that migrate to the tissue from the bone marrow, but it is still not clear whether it functions independently.

**Loss of NLRP12 improves NOD2-driven colonization resistance.** Previous work showed that NOD2 signaling is required for optimal eradication of *C. rodentium*[33,34], which is a mouse-restricted model for attaching and effacing (A/E) enteric bacterial-induced diarrhea such as those caused by EPEC and EHEC in humans. *C. rodentium* colonizes the caecum and the colon of mice through attachment to the epithelium, effacement of microvilli-covered surface and the formation of pedestal-like structure[9]. Given that NLRP12 negatively MDP tolerance by regulating the stability of NOD2/RIPK2 complex, we next determined whether loss of NLRP12 may protect mice from *C. rodentium* as a potential consequence of a greater epithelial expression of IFIT2[11,34]. To this end, we orogastrically inoculated wild-type, *Ifit2*-deficient and *Nlrp12*-deficient mice with $1 \times 10^9$ colony-forming units of *C. rodentium*. A kanamycin-resistant strain of *C. rodentium* was used for non-invasive monitoring of bacterial growth in the feces. As expected[35], a typical bacterial shedding curve was observed in wild-type mice, however the intestine of *Nlrp12*-deficient mice was less rapidly colonized by *C. rodentium* (Fig. 6a). In contrast, the peak of bacterial colonization occurred earlier in *Ifit2*-deficient mice (Fig. 6b) as was observed in the absence of STAT1 that is required for Caspase11-dependent inflammasome activation in response to *C. rodentium*[36]. The latter results suggested to us that enhanced IFIT2 secretion at steady state might protect the intestine of *Nlrp12*-deficient mice against *C. rodentium*, which is most likely caused by a loss of tolerance to MDP. We next compared the colonization dynamic of *C. rodentium* in mice that are deficient for both NLRP12 and NOD2. Similar to *Nod2*-deficient mice (Supplementary Fig. 17a), bacterial burden among compound mutant mice was persisting even 2 weeks post-infection (Fig. 6c). Consistent with a tolerogenic property of NLRP12 on NOD2 signaling in monocytes, intraluminal elimination of *C. rodentium* was improved in lethally irradiated mice that were specifically engrafted with the bone marrow cells of *Nlrp12*-deficient mice although this difference failed to reach statistical significance (Fig. 6d). Conversely, the colonization resistance was found to be similar in both control

and mutant mice that were reconstituted with wild-type bone marrow cells (Fig. 6e). To obtain additional evidence that such colonization resistance result from a loss of MDP tolerance in monocytes, controls and LysM-Cre;*Nod2*<sup>fl/fl</sup> mice, in which *Nod2* gene is specifically deleted in LysM-expressing cells, were orally infected with *C. rodentium* for analyzing the intraluminal bacterial load. A similar number of the pathogen to what seen in *Nod2*-deficient mice was recovered from feces of LysM-Cre; *Nod2*<sup>fl/fl</sup> mice compared to *Nod2*<sup>fl/fl</sup> animals and to those that express the Cre recombinase in IECs (Supplementary Fig. 17b). Such delayed clearance of *C. rodentium* from the gut lumen resulted in a greater bacterial dissemination in the spleen (Supplementary Fig. 17c). This was associated with spenomegaly (Supplementary Fig. 17d) and tissue pathology as evidenced respectively by about 47 percent increase in spleen weight (Supplementary Fig. 12d) and by the enhanced histological score (Supplementary Fig. 17e) and a 30 percent increase in the crypt length (Supplementary Fig. 17f-g). To further examine the basis of how NOD2-mediated host defense may protect *Nlrp12*-deficient mice against *C. rodentium*, RNA was extracted from the caecum of infected mice at day 0 and day 7 post-infection and a genome-wide analysis of the acute transcriptional response to the pathogen was performed (Supplementary Data 1). As expected, *C. rodentium* infection was found to robustly induce expression of many genes involved in Th1- and Th17-mediated host defense[37]. We identified a set of 551 transcripts whose expression was modulated at least one log$_2$-fold change in response to the infection independently of NLRP12 expression (overlapping zone in Venn diagram; Supplementary Fig. 18a–b). By contrast, the expression of 427 genes showed a greater degree of change as a consequence of NLRP12 deficiency. Gene ontology analysis of these 427 differentially expressed genes revealed a significant enrichment of up-regulated molecules that were primarily assigned to the functional categories of T helper cell differentiation and granulocyte adhesion and diapedesis (Supplementary Fig. 18c–f). In agreement with these results, the colon lengths of *Nlrp12*-deficient mice was significantly shortened at day 7 post-infection when compared to controls (Fig. 6f). An enhanced luminal level of lipocalin-2 that is primarily secreted by neutrophils was subsequently found in the absence of NLRP12 as early as day 8 after bacterial infection (Fig. 6g). This was correlated with a greater thickening of the colonic mucosa (Fig. 6h) and with an increased length of the colonic crypts of *Nlrp12*-deficient mice (Fig. 6i) that is most likely caused by greater recruitment of leukocytes in response to the pathogen. To further explore this possibility, the mononuclear phagocyte system from the colon of mice was examined in response to the pathogen (Supplementary Fig. 19). Cytofluorometry analysis confirmed enhanced accumulation of Ly6C<sup>hi</sup>MHCII<sup>hi</sup>CCR2<sup>+</sup> cells within cell suspensions of the colonic *lamina propria* of infected *Nlrp12*-deficient mice when compared to that in controls on day 4 (Fig. 6j, k and Supplementary Fig. 20). When compared to that in *Nlrp12*<sup>−/−</sup>:*Nod2*<sup>−/−</sup> mice, the colon of *Nlrp12*-deficient mice showed an increased accumulation of macrophages (265 ± 62 vs. 54 ± 49 of Ly6C<sup>-</sup>CD64<sup>+</sup>CD11b+ cells among a pool of 10,000 Lin<sup>-</sup> MHCII<sup>+</sup> mononuclear cells) (Fig. 6l, m) in which NLRP12 has been shown to negatively regulate extracellular signal-regulated kinases 1 and 2 (ERK1/2) signaling[38]. This was followed by an increased phosphorylation of ERK1/2 as what observed in response to *Salmonella enterica* serovar Typhimurium[6] (Supplementary Fig. 21a). Such imbalance in the tolerogenic milieu suggested to us that *C. rodentium* may exploit NLRP12 signaling for limiting the accumulation of monocytes when being recruited at the site of the infection. By contrast, no change in lipidation of LC3 was observed at either day 7 or 14 post-infection (Supplementary Fig. 21b) and the monocytes isolated from intestine of *Ifit2*-deficient mice showed a similar MHCII expression with a progressive loss of Ly6C marker when compared to that in control mice (Supplementary Fig. 22).

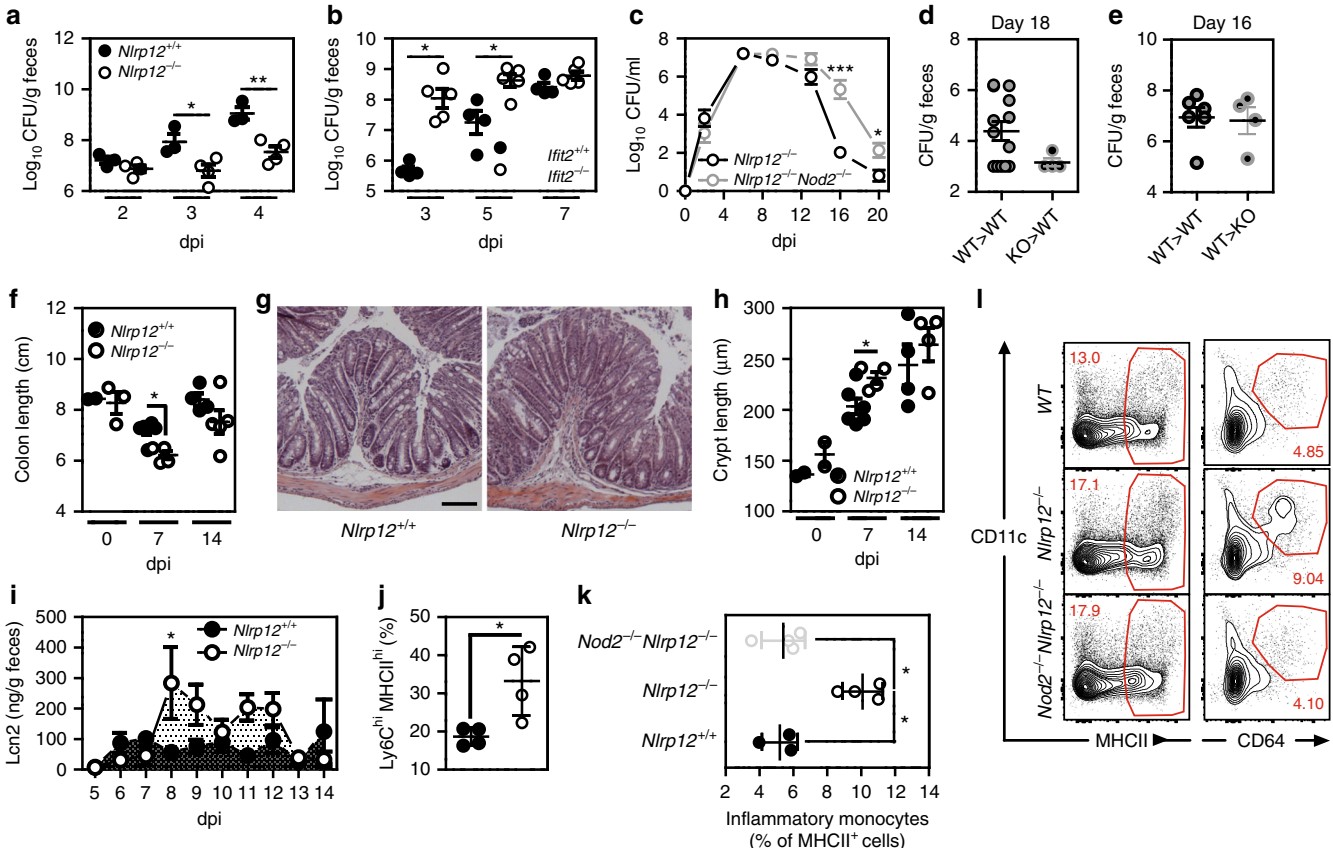

**Fig. 6** NOD2 signaling in monocytes confers protection of *Nlrp12*−/− mice against *C. rodentium* infection. Bacterial-driven colitis was induced by oral gavage of 1x10⁹ *C. rodentium*. **a** *Nlrp12*−/− mice (white circles) and wild-type (black circles) controls on days 2, 3, and 4. **b** CFU counts of *C. rodentium*-infected *Ifit2*−/− (gray circles) and wild-type (black circles) mice on days 3, 5, and 7. **c** Whole infection course was monitored during 3 weeks by calculating CFU per mL of fecal suspensions from *Nlrp12*−/− (*n* = 7) and *Nlrp12*−/−*Nod2*−/− (*n* = 6) mice. **d**, **e** Pathogen burden in feces from lethally irradiated wild-type and *Nlrp12*−/− recipient mice that were reconstituted with bone-marrows from either wild-type or mutant mice. **f** Colon lengths, **g** Representative H&E staining of 5 μm-thick tissue sections from distal colon of *Nlrp12*−/− and wild-type mice on day 7 post infection. Scale Bar represents 200 μm. **h** Inflammatory response of *Nlrp12*−/− and control mice to *C. rodentium* infection was evaluated by calculating crypt lengths from distal colon, **i** LCN2 protein in feces of infected mice. **j** Mean proportion of Ly6Chi MHCIIhi subset as a percentage of live Ccr2-expressing colonic CD11b⁺ cells from the lamina propria of chimeric mice. **k** Mean proportion of inflammatory monocytes as a percentage of live MHCII-expressing colonic CD11b⁺ cells from the lamina propria of wild-type and mutant mice. **l** Representative expression of CD11c and CD64 expression amongst total live fraction of MHCII-expressing colonic CD11b⁺ cells from *Nlrp12*−/−, *Nod2*−/− *Nlrp12*−/− and wild-type mice. Statistical significance was assessed by non-parametric Mann–Whitney test. *P* < 0.05 (*) and *P* < 0.01 (**) were considered statistically significant. Data represent mean ± SEM

Altogether, these results demonstrated that NOD2 signaling promotes accumulation of phagocytes to the site of infection in *Nlrp12*-deficient mice, which may subsequently contribute to the improved clearance of *C. rodentium*[39].

## Discussion

Herein, we unveil a function of NLRP12 as a checkpoint blocker of NOD2 signaling in monocytes that is responsible for the accumulation of inflammatory monocytes. Mutational analysis led us to identify the essential domain (residues 200–224) of NLRP12 that is interacting with the Crohn's disease predisposing NOD2 protein. Consistent with these findings, our finding indicated that the nucleotide-bound state of NLRP12 may repress the stability of NOD2 by sequestering HSP90 that is required for stabilizing the NOD2/RIPK2 complex in response to MDP[16]. This highly conserved region of NLRP12 may contribute to some conformational changes of HSP90. It shares similarities with the fish-specific NACHT associated domain (referred as PF14484) and is responsible for the degradation of NIK[2], which is known to regulate the NOD2-mediated signaling pathway in monocytes[40].

Noteworthy, it includes the ATP/GTP-specific phosphate binding loop called Walker A and a Mg2+ coordination site called Walker B. This agrees with previous reports indicating a suppressive function of ATP-binding domain on the activity of the plant R protein I-2 that shows similarities with NLRP12[41]. As a consequence of its interaction with NLRP12, NOD2 was ubiquitinated and the NOD2/RIPK2 complex showed lowered stability in vitro. Such non-resolving inflammatory conditions result in protective immunity against an enteric bacterial pathogen, although it is unclear whether this reflects an improved NOD2-dependent survival of newly extravasated blood monocytes. Further, we provide evidence that NLRP12 signaling within the hematopoietic compartment is exploited by *C. rodentium* in mice as what observed with *Salmonella enterica* serovar *Typhimurium*[34]. This agrees with a recent report showing enhanced NOD2-mediated protective immune response against *C. rodentium* in animals that are hypomorphic for Atg16L1 protein expression[15]. Such paradigm may reflect a potential selective advantage of disease-causing NLRP12 mutations against some enteric bacterial infections as a consequence of a loss of MDP

tolerance. Thus, our studies could help to explain the recurrent episodes of serosal inflammation (including peritonitis and abdominal pains) that are triggered by exposure to cooling temperatures or cold in patients bearing non-sense mutations in the NLRP12-encoding gene[1]. Ultimately, our findings could lead to new strategies for improving disease tolerance in patients with NLRP12 mutations through modulation of NOD2-mediated control of the intestinal permeability and the gut microbiota[42] that are regulating the body's energy expenditure. Indeed, changes in the composition of the gut microbiota contribute to the susceptibility of colitis that is caused by NLRP12 deficiency[43]. Alternatively, one may also consider that loss of NLRP12 may influence the persistence of some viruses (e.g., Norovirus) as a consequence of the greater epithelial expression of several ISG, including IFIT2 that is involved in antiviral immunity and OAS2 as a RNA-binding protein that interacts with NOD2[44]. Thus, the exact role of NLRP12 signaling on NOD2-mediated resilience of the gut microbiota now deserves further experimental studies with littermate controls and co-housed mice. In this context, we are currently investigating the cause of such influx of inflammatory monocytes that correlated with ISG induction within the epithelium of *Nlrp12*-deficient mice. Collectively, we provide the rational for targeting therapeutically NOD2 signaling in familial cold auto-inflammatory syndrome by unveiling an unappreciated molecular link with the pathogenesis of Crohn's disease.

## Methods

**Mice.** *Nlrp12*-deficient (*Nlrp12*$^{-/-}$) mice were generated through homologous recombination by using the Lex-1 ES cells that are derived from the 129SvEvBrd strain (Supplementary Fig. 9a). A gene-targeting vector with a neomycin-resistance cassette was constructed to replace the first two exons of *Nlrp12*. The latter is required to encode the Pyrin domain of NLRP12, which is essential for recruiting ASC and subsequently for activating Caspase-1. Genotyping of positive ES clones was accomplished by Southern blotting analysis. *Nlrp12*$^{-/-}$ mice were produced at the expected Mendelian ratio by crossing heterozygous animals and were crossed with *Nod2*-deficient mice[12] for generating animals that are deficient for both NOD2 and NLRP12. Genotyping of mouse tail DNA was performed to confirm the presence of the wild-type and/or targeted alleles (Supplementary Fig. 9b). The absence of *Nlrp12* mRNA in *Nlrp12*$^{-/-}$ animals was confirmed by quantitative reverse-transcriptase (RT)-qPCR (Supplementary Fig. 9c). *Nlrp12*-deficient (*Nlrp12*$^{-/-}$) mice were backcrossed onto a C57BL6/J background. *Ifit2*-deficient (*Ifit2*$^{-/-}$) were generated as described elsewhere and backcrossed onto a C57BL/NCrl background (N10)[29]. All animal studies were approved by the local investigational review board. Animal experiments were performed in an accredited establishment (N° B59-108) according to governmental guidelines N°86/609/CEE. Age- and gender-matched animals were housed up to five per cages and had free access to a standard laboratory chow.

**Bacterial infection.** Age and sex-matched mice were orally inoculated with ~1 × 10$^9$ CFU of either *Citrobacter rodentium* strain DBS100 or kanamycin-resistant *C. rodentium* strain DBS120 for colony forming unit (CFU) counting in feces (kindly provided by D. Schauer, Massachusetts Institute of Technology). Histological scoring of inflammatory cells infiltration and of crypt length damage was blindly performed on hematoxylin and eosin (H&E) stained sections by two investigators.

**Bone marrow transplantation experiments.** Recipient mice underwent a lethal total-body irradiation. Twenty-four hours post-irradiation, mice received intravenously 5 × 10$^5$ fresh bone marrow cells. Blood was collected in heparin-containing tubes 7–8 weeks after bone-marrow transplantation and reconstitution efficiency was checked by flow cytometry using a FACS Canto II (BD biosciences) after cell staining by PE- conjugated anti-CD45.1 (A20) and FITC- conjugated anti-CD45.2 (104) from BD biosciences. Two months after bone-marrow transplantation, the colonization resistance towards *C. rodentium* and the waterfall-shaped flow cytometric distribution of monocyte descendants was analyzed within the colon of chimeric mice.

**Septic shock.** Male mice (8–10 weeks old) were injected intraperitoneally with a non-lethal dose of highly purified LPS (10 mg/kg of highly purified *E. coli* 0111:B4 purchased from Invivogen) 24 h before a secondary challenge with murabutide at 10 mg/kg (Invivogen). Alternatively, mice were challenged with a lethal dose of highly purified LPS as a model of acute endotoxin septic shock (54 mg/kg of highly purified *E. coli* 0111:B4 purchased from Invivogen). Mice were monitored twice

daily over a 6-day period. The morphology of recruited cells within the peritoneum of mice was determined by cytological examination after centrifugation on glass microscope slides (Cytospin; Shandon), fixation and staining according to the manufacturer's recommendations (Diff; Dade Behring Inc.). For cytokine measurements, serum was taken at 90, 180, 360, and 540 min after secondary MDP challenge.

**Immunoprecipitations.** In all, 2.5 × 10$^6$ HEK293T cells were transfected with the indicated constructs using Lipofectamine2000 (Invitrogen), as indicated in the manufacturers' protocol. Cells were harvested in RIPA buffer 24 h post transfection (150 mM NaCl, 50 mM Tris pH 7.4, 1% Triton X-100, 0.1% SDS, 0.5% Na-deoxycholate) containing phosphatase inhibitors (20 M β-glycerophosphate, 5 mM NaF, 100 mM Na$_3$VO$_4$) and protease inhibitors (Complete protease inhibitor cocktail with EDTA; Roche). The lysate was cleared 20 min at 14,000 × *g* for 20 min and FLAG-tagged NOD2 was subsequently precipitated for 3 h at 4 °C using anti-FLAG M2 agarose (Sigma-Aldrich). Proteins were separated by Laemmli SDS-PAGE and visualized using either anti-FLAG M2 mouse antibody (Stratagene, #200471, 1:2000) or anti-Myc mouse antibody 9E10 (Roche, Cat. No. M4439, 1:1000) or anti-cMYC rabbit antibody (Santa Cruz, sc-789, 1:1000).

**CRISPR/Cas9 gene editing.** The NOD2 (NP_001280486) and NLRP12 (NM_001277126) loci were deeply examined; a common translational start for all the reported isoforms were selected and used to generate KO CRISPR guide RNA pairs. The gRNAs were subsequently identified using the Sanger Centre CRISPR webtool (http://www.sanger.ac.uk/htgt/wge/find_crisprs). The chosen gRNAs were designed to cut as far upstream as possible to generate indels in the region containing the ATG start codon; an additional G were added to the 5′ end of the guides to maximize expression from the U6 promoter. Complementary oligos were designed and annealed to yield dsDNA inserts with compatible overhangs to BsmBI-digested vectors[45], the sense guides were inserted into the puromycin selectable plasmid LentiCRISPR/Cas9v2 (Addgene #52961). HEK293FT cells were co-transfected with the appropriate Lentivirus plasmids. Following 24 h of recovery and a further 48 h of puromycin selection (2 µg/mL), cells were subjected to a further round of puromycin selection to enrich for transfectants. THP-1 cells were infected with the produced lentivirus harboring the gRNA and the Cas9 protein. The cell pools were subsequently single cell sorted by FACS and clones analyzed for NLRP12 or NOD2 depletion by immunoblotting and sequenced. Briefly, genomic DNA was isolated from cell candidates and the region surrounding the ATG start codon of both NOD2 and NLRP12 were amplified by PCR using a forward and a reverse primer (i.e., indel primers). The resulting PCR products were subcloned into the holding vector pUC19 and around 10 colonies were picked for each clonal line. Plasmid DNAs were isolated and sent for sequencing with primers M13F and M13R to finally select a cell line. The absence of the NOD2 or NLRP12 protein was confirmed by western-blotting.

**Generation of THP1 cells stably expressing NOD2.** The stable THP-1 cell line was generated using a retroviral system. The Myc-BirA*-NOD2 was constructed using the pLXN-retroviral vector (ClonTech, USA). The retrovirus vector was transfected into HEK293-FT cell line for the production of viruses. The viruses were harvested after 72 h and the THP-1 cells were infected and subsequently selected using 250 µg/mL of Geneticin® (LifeTechnologies, USA). After 14 days, the surviving cells were cell sorted and individual clones were grown for 2 months. The individual clones were tested for the proper expression of Myc-BirA*-NOD2 using the anti-NOD2 (2D9) monoclonal antibody (SantaCruz, sc-56168, 1:250).

**Luciferase reporter assays.** The NLRP12 coding sequence was inserted into the pcDNA3.1 vector and site-directed mutagenesis was performed to generate the plasmids expressing the Arg284X and Arg352Cys mutations. 3 × 10$^4$ HEK293T cells were seeded in a 96-well format directly prior transfection with the indicated amounts of plasmid using XtremeGene9 (Roche) as indicated in the manufacturers' protocol. Cells were harvested in luciferase lysis buffer 24 h post transfection (25 mM Tris pH 8, 8 mM MgCl$_2$, 1% Triton, 15% glycerol, 1 mM DTT) and luciferase activity was measured using a standard plate luminometer (Berthold Instruments). Luciferase activity was normalized as a ratio to β-galactosidase activity and standard deviation (SD) was calculated from triplets.

**Analysis of human peripheral mononuclear cells.** Patients with hereditary recurrent fever carrying the non-sense c.850 C>T (p.R284X) heterozygous mutation in NLRP12 were previously described[1]. The blood from patients and controls was collected in citrate collection tubes. PBMCs were isolated by centrifugation using Ficoll-Paque (GE Healthcare Life Sciences). Fresh cells were stimulated with Ultrapure LPS (10 ng/mL; Sigma-Aldrich), MDP (10–100 µg/mL; Invivogen), MDP-DD (10–100 µg/mL; Invivogen), or left unstimulated for 24 h. Supernatants were collected for ELISA analysis. The treatment with 30 µg/mL of cycloheximide (Sigma-Aldrich) for 2 h was performed on 5 million PBMCs that were isolated from fresh blood of the two affected patients using Pancoll gradient centrifugation (BioTech) and cultured in RPMI 1640 medium.

**Flow cytometry analysis**. Cells were stained and analyzed using a FACS LSRFortessa™ system (BD Biosciences). Dead cells were excluded with the LIVE/DEAD Fixable Violet Dead Cell staining kit (Life technologies). Lineage-positive cells were excluded using the PerCP5.5-conjugated anti-CD3 (17A2), anti-NK1.1 (PK136), anti-CD19 (6D5), anti-Ly6G (1A8) (Biolegend). PerCP-conjugated anti-CCR3 (83103) added to the lineage staining to exclude eosinophils was from R&D. Allophycocyanin-Cy7-conjugated anti-CD11b (M1/70), PerCP5.5-conjugated anti-Ly6G (1A8), Brilliant violet 510-conjugated anti-MHC Class II (I-A/I-E) (M5/114.15.2) and FITC-conjugated and Alexa Fluor 700-conjugated anti-Ly6C (AL21) were from BD Pharmingen. Allophycocyanin-conjugated anti-CD11c (HL3), PE-conjugated and APC-conjugated anti-CD64 (X54-5/7.1), Alexa-Fluor 700-conjugated anti-MHC Class II (I-A/I-E) (M5/114.15.2), PECF594-conjugated anti-CD11c (HL3), Brilliant Violet 570-conjugated anti-Ly6G (1A8), Allophycocyanin-conjugated anti-CD64 (X54-5/7.1), PE Cy7-conjugated anti-CD24 (M1/69), Brilliant violet 650-conjugated anti-CD45.2 (104) and Brilliant violet 711-conjugated anti-CD45.1 (A20) were all from Biolegend. PE-conjugated anti-CCR2 (475301) was from R&D systems. The data were analyzed using the FlowJo software (Tree Star).

**Cytokine measurement**. Cytokine levels were determined by ELISA kits, according to protocols provided by R&D Systems.

**Microarray and gene-ontology analysis**. Caecum specimens from non-infected and infected wild-type and $Nlrp12^{-/-}$ animals were dissected out and stored at −80 °C in RNAlater® (Ambion, Applied Biosystems, Foster City, CA), until extraction of total RNA accordingly to manufacturer's instructions (Qiagen). The quality of the extracted RNA was confirmed by Agilent 2100 Bioanalyzer using RNA Nano 6000 (Agilent Technologies). The 4x44K Whole Mouse Genome Oligo Microarrays (Agilent Technologies) was used to determine the gene expression profile of two biological replicates. For each labeling, 2 μg of total RNA per sample were engaged in the synthesis of a fluorescent probe labeled with Cy5 or Cy3 fluorophores. A 2 × 2 factorial experimental design and a dye-swap strategy were used (GEO accession number GSE59940). After a within array loess normalization, raw data were analyzed with the LIMMA package and sets of differentially expressed genes were filtered for a *p*-value <0.01 and a limit log fold change >1 by using moderated *t*-statistic with empirical Bayes shrinkage of the standard errors. Statistics were corrected for multiple testing using False Discovery Rate approach. A gene-ontology analysis using Panther was next performed on up- and down-regulated genes that are referred in Unigene (http://www.pantherdb.org/).

**Gene expression analysis**. Isolated RNA was reverse-transcribed with the High-Capacity cDNA Archive kit (Applied Biosystems), according to the manufacturer's instructions. The resulting cDNA (equivalent to 5 ng of total RNA) was amplified using the SYBR Green real-time PCR kit and detected on a Stratagene Mx3005P (Agilent Technologies). RT-PCR was performed with the forward and reverse primers (sequences available upon request) that were designed using Primer express software, version 1.0 (Applied Biosystems, Foster City, CA). On completion of the PCR amplification, a DNA melting curve analysis was carried out in order to confirm the presence of a single and specific amplicon. *Actb* was used as an internal reference gene in order to normalize the transcript levels. Relative mRNA levels ($2^{-\Delta\Delta Ct}$) were determined by comparing (a) the PCR cycle thresholds (Ct) for the gene of interest and *Actb* (ΔCt) and (b) ΔCt values for treated and control groups (ΔΔCt). RNA extraction from whole peripheral blood from controls and FCAS2 patients that were collected in PAXgene tubes was performed using PAXgene Blood RNA Kit following the manufacturer's instructions (Qiagen). RNA extraction from PBMCs was performed using the RNeasy Mini Kit (Qiagen) including DNase treatment according to the manufacturer's instructions. One μg of RNA was reversed transcribed in the presence of 2.5 mM of oligo-dT using the Reverse Transcriptor kit (Roche) following the manufacturer's instructions. 75 ng of cDNAs were amplified using Q5 High-Fidelity 2X Master Mix (New England BioLabs). Forward and reverse primers used in PCR amplification are located in the 5'UTR and 3'UTR of NLRP12 cDNA respectively (Sequences available upon request), which can amplify the two alleles of NLRP12 in the patients and the healthy donor. Exon 3 of NLRP12 cDNA was sequenced with the Big Dye Terminator sequencing kit (Applied Biosystems) using two different primers (sequences available upon request), and run on an ABI 3730 × l automated sequencer. Sequences were analyzed with SeqScape software (Applied Biosystems).

**Assembling of K48 poly-ubiquitin chains on NOD2**. Assembling of K48 poly-ubiquitin chains on NOD2 in response to MDP was monitored over time. THP-1 Myc-BirA*-NOD2 cells were pre-treated for one h with DMSO (vehicle) or MG132 (at 12.5 μM) before 10 μg/mL of MDP (InvivoGen, France) was added. At given time points, samples were taken, spun down and the reaction was stopped using RIPA lysis buffer plus protease and phosphate inhibitors (ROCHE, Germany). Additionally, 50 mM of N-ethylmaleimide (Thermofisher scientific, USA) was added to the RIPA buffer to lyse the MG132-treated samples. A SDS gel was run and membranes were blotted against the anti-NOD2monoclonal antibody (2D9) (SantaCruz, sc-56168, 1:250) and RIPK2 (Cell Signalling Technology, USA) and anti-actin (Sigma Aldrich, UK, Cat No. A5060, 1:2000) or anti-Tubulin (Sigma Aldrich, UK, Cat No. T9026, 1:2000) as a controls. NOD2 ubiquitination was assessed using the anti-poly-K48 antibody (Merck Millipore, USA, Cat No. 05-1307, 1:2500).

**Ubiquitination analysis**. The THP-1 Myc-BirA*-NOD2 stable cell line was grown and a total of $1 \times 10^7$ of cells were used for these experiments. Cells were incubated for 2 h with MDP (10 μg/mL). After the 2 h incubation with and without proteasome inhibitor MG132 (12.5 μM), the cells were placed on ice, washed twice with PBS, spun down and incubated in 1 mL of lysis buffer (50 mM HEPES pH 7.5; 150 mM NaCl; 1× complete-EDTA free protease inhibitor; 1X phosphatase inhibitor; 1% iGPAL and 1 mM PMSF). Samples were incubated for 10 min and then spun down at $14,000 \times g$ for 20 min at 4 °C. The Supernatant was separated and incubated with anti-Myc magnetic beads (Thermofisher, UK) for 3 h. Beads were washed 5 times using the lysis buffer before they were eluted with a 0.1 M Glycine solution pH 2.0. Finally, 15 μL were loaded onto the gel and immunoblotted using the monoclonal anti-poly-K48 antibody (Merck Millipore, USA, Cat No. 05-1307, 1:2500) or the monoclonal anti-NOD2 (2D9) antibody (SantaCruz, sc-56168, 1:250).

**Statistics**. Data were analyzed using Prism4.0 (GraphPad Software, San Diego, CA). The non-parametric Kruskal–Wallis test with Dunn's multiple comparison test or the parametric one-way ANOVA test with Bonferroni's multiple comparison test were used. Values represent the mean of normalized data ± SEM. Asterisk, significant difference $P < 0.05$.

## Data availability

The microarray data have been deposited under GEO accession number GSE59940. A reporting summary for this Article is available as a Supplementary Information file.

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

## Acknowledgements

We thank Yvonne Postma for technical assistance.

## Author contributions

S.N. and N.W. performed most of the experiments. C.C., A.C-M., O.B., and C.B. supported the mouse experiments. A.N. and J.M-T. performed co-immunoprecipitation experiments and/or NF-kappaB assays with the help of K.L., R.R., and A.R-R. M.D. performed immunoblot analysis on intestinal tissues and Lipocalin-2 measurements in feces. A.C-M. performed bone marrow chimera experiments with the help of C.C. and N. W. C.C. and L.F.P. performed FACS analysis with the help C.S. and M.G. L.H. and D.H. performed and analyzed microarray analysis. Studies on patients' cells were performed by C.C., L.C., and F.A. with the help of S.K. and S.A. A.S., E.B., and B.R. supplied novel cell lines and/or transgenic mice. M.C. conceived and supervised the study. M.C., T.A.K., and L.F.P. analysed the data and wrote the manuscript.

## Additional information

**Competing interests:** The authors declare no competing interests.

