## [Peer Review File · Nature Communications]

Reviewers' comments:

Reviewer #1 (Remarks to the Author):

In the manuscript “Proteosomal degradation of Nod2 by Nlrp12 promotes bacterial tolerance and colonization by pathogen” the authors describe a new role for NLRP12 as a negative regulator of NOD2 activity and expression. The authors confirm the finding of a previous report that NLRP12 associates with NOD2 and further show that NLRP12 expression promotes NOD2 ubiquitination. A functional role for this mechanism is further described in experiments showing Nlrp12-deficient mice are more susceptible to MDP challenge following low-dose LPS exposure, and *Citrobacter rodentium*-infected Nlrp12-deficient mice display a NOD2-dependent hyperinflammatory response and reduced bacterial colonization of the intestine. Although the authors present convincing support that NLRP12 suppresses NOD2 function and provide elegant and clinically-relevant results using the human disease-associated R284X NLRP12 mutant, additional controls and experiments would greatly cement the mechanistic link between the NLRP12-NOD2 relationship and provide better cohesion between the major findings of the manuscript.

Major Points

1. In Figure 1, the authors overexpress NOD2 and NLRP12 truncation mutants in 293T cells to confirm that these proteins interact and to determine which NLRP12 domains facilitate the NLRP12-NOD2 interaction. The authors conclude “these results suggested that NLRP12 may interfere with NOD2 signaling in myeloid cells where NLRP12 is primarily expressed”. Although the authors later use bone marrow chimeras to show NLRP12 expression by myeloid cells regulates interferon-stimulatory genes (ISGs) in the intestine, the authors should also demonstrate that NLRP12 and NOD2 can interact in myeloid cells or more importantly, demonstrate that the presence of NLRP12 alters NOD2 expression and activity in myeloid cells (similar to the data presented in Fig. 2).

2. In Figure 2, the authors show co-expression of NLRP12 with NOD2 leads to reduced NF- κ B (p50/p65) reported activity induced by MDP. The authors further

show NLRP12 promotes NOD2 ubiquitination and reduces NOD2 protein expression in cycloheximide-treated 293T cells, suggesting the loss of NOD2 activity is due to proteasome-mediated degradation of NOD2 induced by NLRP12. To further demonstrate this, and support the main conclusion of the manuscript, the authors should repeat these assays in the presence of proteasome inhibitors (e.g., MG-132) to determine if this prevents NOD2 degradation and returns NOD2 activity in the presence of NLRP12.

In addition, the authors should consider if IRAK hyperphosphorylation is altered in MDP-stimulated cells co-expressing NLRP12 and NOD2 since MDP can activate IRAK and NLRP12 restricts NF- κ B partly via suppressing IRAK hyperphosphorylation; thus, NLRP12 may limit NOD2 activity by suppressing IRAK.

3. In figure 3, the authors elegantly screen disease variants of NLRP12 and show that the NLRP12 R284X mutant fails to suppress NOD2-dependent NF- κ B activation, and human PBMCs with the NLRP12 mutation have enhanced IL-1 β production in response to MDP. While these studies provide important clinical relevance to the authors main conclusion that NLRP12 acts as a negative regulator of NOD2, there is some confusion as to how the R284X mutant fails to prevent NOD2 activity, while still retaining the ability to associate with NOD2 (Fig. 3d). The authors should reconcile this by determining if the NLRP12 mutant fails to induce NOD2 ubiquitination, similar to the experiment presented in Figure 2a.

In addition, the protein levels of the R284X mutant appears to be undetectable in the western blots shown in Fig. 3c-d, which suggest the possibility that this mutant fails to suppress NOD2 because it is no longer expressed. This would conflict with the authors results presented in Fig. 3d and conclusion that this mutant associates with NOD2. Alternatively, is the R284X mutant reduced in size, as suggested by the immunoblot where the authors immunoprecipitated and then immunoblotted NLRP12 (Fig. 3d, lower immunoblot, 4th lane)? Can the

protein levels of the R284X mutant only be detected when first immunoprecipitated (opposed to immunoblotting the inputs)? Regardless, the authors should address this concern more thoroughly.

4. In figure 5, the authors generate bone marrow chimeras to determine the cellular compartment expressing NLRP12 that is responsible for regulating ISGs in the intestine and provide support for a myeloid expression of NLRP12 in regulating this. Although this is consistent with previous reports describing the role of NLRP12 in inflammatory bowel disease and colitis-associated colon cancer, the authors should provide further functional assessments of myeloid vs. non-myeloid NLRP12 in vivo by assessing *C. rodentium* colonization in bone marrow chimeric mice, which would better support the overall conclusion of the manuscript, which focuses on bacterial tolerance and colonization.

5. The authors demonstrate that *Nlrp12^{-/-}/Nod2^{-/-}* double deficient mice have increased bacterial burdens compared to wild type mice (and *Nlrp12^{-/-}* single-deficient mice based on Fig. 6c). It would be experimentally challenging to measure NOD2 ubiquitination in *Nlrp12*-sufficient and deficient mice during *C. rodentium* infection; however it would be important to determine if NLRP12 regulates NOD2 protein expression in vivo to fully support the claim that NLRP12 promotes NOD2 degradation to regulate bacterial colonization. In addition, the authors need to measure NF-κB activation in the colons of *C. rodentium*-infected mice as a measure of down-stream NOD2 activation.

Minor Points

In figure 1b, RIPK2 and NOD2 expression are extremely low or undetectable in the “input” immunoblots, despite both of these proteins being overexpressed and easily visible in the “IP” immunoblots. A darker exposure needs to be included to confirm expression and to accurately compare the IP blots.

In the results section, the figures 3c+3d are switched.

Please add labels to the bar graphs shown in Figure 5b to clarify which experimental group these graphs are referring to.

On page 10, lines 287-288, the authors state “we next assessed whether the transcriptional response of the intestine of mutant mice may potentially confer a growth advantage to *C. rodentium*” despite showing that *C. rodentium* is cleared more efficiently in *Nlrp12*-deficient mice in the previous figure (Fig. 6c). It seems the hypothesis should be framed in the opposite direction to fit with the presented data.

The authors describe colon crypt lengthening (hyperplasia) as an indicator of disease in Fig. 5. Both hyperplasia and immune infiltrate can be semi-quantitatively scored in IHC-stained colons by a trained pathologist. This would allow the authors to determine if significant differences are present for these disease parameters. Similarly in Figure 6e and supplementary figure 8, a shortened colon and amplified ERK1/2 pathways were shown which usually indicate an increased intestinal inflammation. Do the authors have data on weight loss observation and histopathological score for the intestine to indicate colitis?

In figure 7g, do the authors have growth curves for *C. rodentium*-infected *Nlrp12*^{-/-} mice to better directly compare these mice to the infected wild type and *Nlrp12*^{-/-}/*Nod2*^{-/-} double deficient animals, similar to the data presented in Fig. 7h-j.

Reviewer #2 (Remarks to the Author):

In the current manuscript, authors demonstrated a role of NLRP12 in NOD2 signaling and bacterial tolerance using cells from FCAS2 patients and NLRP12-KO mice. This study is interesting in terms of the extension of role of NLRP12. However, there are several points are required to be addressed by authors.

1. Overall flow of description in this study. In the beginning, authors showed data from transfection study showing NLRP12-NOD2 connection. Actually authors showed suppression of NOD2 degradation and NF-kB activation. Then authors demonstrated FCAS2 patients and various mouse models. In FCAS2 patients and NLRP12-KO mice, authors focused on ISGs without showing the degradation of NOD2 and NF-kB activation, which should be provided for convincing story.
2. Figure 1. Demonstrated interaction of NLRP12 with NOD2 through linker region of NLRP12. But the construct without PYD-Linker also interacts (Fig 1c), which suggested involvement of multiple domains for the interaction.
 - (a) Thus it is needed to show which remaining domains (NBD or LRR) are involved in the NLRP12-NOD2 interaction.
 - (b) Author should provide endogenous NLRP12 and NOD2 interaction.
 - (c) The quality of Fig.1c is poor: in input blot, the bands are very faint and the amount of NLRP12 is not increased accordingly.
3. Figure 2c, lack of controls: should include NF-kB activation in the presence of various NLRP12 without NOD2.
4. For Figure 3
 - (a) Figure 3a, if the FCAS2 patients are the same patients reported in the previous publication (ref #1), this panel should be removed from main figures and just mention in text with reference.
 - (b) In FCAS patients with R284X, authors insisted haploinsufficiency rather than dominant-negative effect of the mutant NLRP12 (Fig 3f). To prove that, author should show the level of NLRP12 at the endogenous protein level by Western blot in comparing with healthy controls.
 - (c) It is not clear what is the role of Hsp90 on NLRP12 and NF-kB activation. In the previous data, author demonstrated that NLRP12 suppresses NOD2-mediated NF-kB activation through NLRP12-NOD2 interaction and degradation. However, the R284X mutant NLRP12 is still binding with NOD2 without NF-kB suppression. In contrast, R352C mutant shows decreased binding with NOD2 and suppressing NF-kB activation as much as WT NLRP12. This result is not consistent with previous results. In addition, author should show the level of NOD2 in patients and healthy controls by Western blot whether it is reduced or not because the R284X-NLRP12 mutant probably fails for degrading NOD2.
 - (d) Fig.3c is not appropriate. R284X is smaller than WT or other missense mutant NLRP12. Thus author should show whole range of blot in this figure.
5. Figure 4.
 - (a) High level of LPS challenge results in similar susceptibility but low LPS and MDP challenging shows significant difference. In this experiment, low dose LPS challenging without MDP is missing. This is an important control.
 - (b) Regarding to FCAS2 patients with R284X, which is loss-of-function and haploinsufficiency, author should include heterozygotes (Nlrp12^{+/-}) in this challenging experiments.
 - (c) In this figure, only Cxcl1 levels are provided (Fig. 4a), while FCAS2 patients with R284X has shown increased IL-1 β . Same readings are needed in two related systems (patients and animal model) to make consistent story.
6. Figure 5: What's the NOD2 levels in NLRP2-KO mice? If all of the ISG increases are due to NOD2 activation in NLRP12-KO mice, that should be proved. It is very important in context of whole story of this study.
7. Figure 6: in addition to Fig. 6b (Ifit2-KO) and 6c (Nlrp12-KO), bacterial colonization of double KO for Ifit2 and Nlrp12 is needed to confirm directly that reduced colonization of Nlrp12-KO mice is due to increased Ifit2 expression.
8. In supplementary figure 4, author showed hyper-activation of STAT1, which leads ISG induction. However, in supplementary figure 12, defect of STAT3 is shown. If both STAT1 and 3

are regulated by NOD2 activation. Which signal is important, STAT1 or STAT3, or both? Author should show the status of STAT1 and 3 in three KO mice, NLRP12-, Ifit2-, and NOD2-KO mice.

Minor points:

1. The labeling of Figure 3b and 3c is not match in text (page 6, lines 137 and 139).
2. Legend for Figure 4c is missing.
3. Supplementary table 1 is missing.

Point-by-point response to reviewers' comments

Reviewer #1: *In the manuscript "Proteasomal degradation of Nod2 by Nlrp12 promotes bacterial tolerance and colonization by pathogen" the authors describe a new role for NLRP12 as a negative regulator of NOD2 activity and expression. The authors confirm the finding of a previous report that NLRP12 associates with NOD2 and further show that NLRP12 expression promotes NOD2 ubiquitination. A functional role for this mechanism is further described in experiments showing Nlrp12-deficient mice are more susceptible to MDP challenge following low-dose LPS exposure, and Citrobacter rodentium-infected Nlrp12-deficient mice display a NOD2-dependent hyperinflammatory response and reduced bacterial colonization of the intestine. Although the authors present convincing support that NLRP12 suppresses NOD2 function and provide elegant and clinically relevant results using the human disease-associated R284X NLRP12 mutant, additional controls and experiments would greatly cement the mechanistic link between the NLRP12-NOD2 relationship and provide better cohesion between the major findings of the manuscript*

Author response. We thank the first reviewer for her/his encouraging comments.

1) *In Figure 1, the authors overexpress NOD2 and NLRP12 truncation mutants in 293T cells to confirm that these proteins interact and to determine which NLRP12 domains facilitate the NLRP12-NOD2 interaction. The authors conclude "these results suggested that NLRP12 may interfere with NOD2 signaling in myeloid cells where NLRP12 is primarily expressed". Although the authors later use bone marrow chimeras to show NLRP12 expression by myeloid cells regulates interferon-stimulatory genes (ISGs) in the intestine, the authors should also demonstrate that NLRP12 and NOD2 can interact in myeloid cells or more importantly, demonstrate that the presence of NLRP12 alters NOD2 expression and activity in myeloid cells (similar to the data presented in Fig. 2).*

Author response. We thank this reviewer for pointing this out. Given the poor sensitivity of commercially available NOD2 antibodies, we generated a monocytic THP-1 cell line stably expressing the fusion protein Myc-BirA*-NOD2 by using a retroviral system. This led us to confirm an endogenous interaction of NLRP12 with NOD2 in human monocytic cells when specifically inhibiting the MDP-induced proteasomal degradation of NOD2. These additional data are shown in the novel panel a of the first figure. In line with this finding, the relevance of this interaction is illustrated by a set of additional *in vivo* experiments, that revealed a delayed clearance of

Citrobacter rodentium in mice with specific depletion of Nod2 signaling in myeloid cells (new Fig. 7a). Conversely, a greater elimination of *C. rodentium* was observed in lethally irradiated mice that were specifically engrafted with the bone marrow of *Nlrp12*-deficient mice. The text has been revised accordingly.

2) In Figure 2, the authors show co-expression of NLRP12 with NOD2 leads to reduced NF- κ B (p50/p65) reported activity induced by MDP. The authors further show NLRP12 promotes NOD2 ubiquitination and reduces NOD2 protein expression in cycloheximide-treated 293T cells, suggesting the loss of NOD2 activity is due to proteasome-mediated degradation of NOD2 induced by NLRP12. To further demonstrate this, and support the main conclusion of the manuscript, the authors should repeat these assays in the presence of proteasome inhibitors (e.g., MG-132) to determine if this prevents NOD2 degradation and returns NOD2 activity in the presence of NLRP12. In addition, the authors should consider if IRAK hyperphosphorylation is altered in MDP-stimulated cells co-expressing NLRP12 and NOD2 since MDP can activate IRAK and NLRP12 restricts NF- κ B partly via suppressing IRAK hyperphosphorylation; thus, NLRP12 may limit NOD2 activity by suppressing IRAK.

Author response. We agree with the reviewer that the suggested experiments would add to the strength of the findings. To this end, we have performed a number of experiments to support the main conclusion of our manuscript. Co-immunoprecipitation revealed an accumulation of poly-K48 ubiquitination of NOD2 in response to MDP when THP-1 cells were treated with the proteasome inhibitor MG132 (Fig. 2d). Likewise, the inhibitory effect of NLRP12 on the stability of NOD2 in HEK293T was expectedly compromised upon inhibition of the ubiquitin-proteasome pathway by MG132 (Supplementary Figure 2b). Consequently, the MDP-induced interaction of NOD2 with the chaperone HSP90 was strongly inhibited by the treatment of THP-1 cells by MG132 (Fig. 2e). In contrast, MG132 expectedly failed to promote accumulation of NOD2 and its interaction with HSP90 in THP1 cells when being stimulated by lipopolysaccharide (Fig. 2e). Lastly, we agree with the reviewer that NLRP12 may restrict NF- κ B partly via suppressing IRAK hyperphosphorylation, nevertheless, the endogenous interaction between NOD2 and IRAK1 was barely detectable if any. We then feel that these findings on IRAK-1 can be added in the manuscript if requested. In conclusion, the dedicated paragraph has been entirely rewritten as follow : "To corroborate the role of NLRP12 as a potential checkpoint blocker of NOD2 signaling in monocytes, we examined the influence of NLRP12 on the stability and the activity of the NOD2/RIPK2 complex. Co-immunoprecipitation experiments revealed that NLRP12 expression promotes poly-ubiquitination of

the NOD2/RIPK2 complex in HEK293T cells (Fig. 2a). In contrast, the ubiquitination status of NOD1 was not influenced by full-length NLRP12 (Supplementary Figure 2a). Consistent with a regulation of NOD2 activity at the protein level through its interaction with NLRP12, blocking of protein neosynthesis using cycloheximide led us to reveal that NLRP12 expression reduced the half-life of the NOD2 protein (Fig. 2b). This inhibitory effect of NLRP12 on the stability of NOD2 was expectedly compromised upon inhibition of the ubiquitin-proteasome pathway by MG132 (Supplementary Figure 2b). Accordingly, transient expression of full-length human NLRP12 greatly reduced the NOD2-mediated p50/p65 reporter activation in response to MDP in HEK293T cells (Fig. 2c), while increasing amounts of transfected NLRP12 further reduced MDP-induced NF- κ B activation by about 70% (Fig. 2c) paralleling the lowered protein levels of both NOD2 and RIPK2 (Fig. 1b). Equally of importance, the endogenous formation of the protein complex between NOD2 and NLRP12 coincided with a greater assembling of K48 poly-ubiquitin chains on NOD2 when pretreating THP1 cells stably expressing the fusion protein Myc-BirA*-NOD2 with the proteasome inhibitor MG132 (Fig. 2d). As reported previously¹⁹, co-immunoprecipitation experiments confirmed that NOD2 endogenously interacts with the chaperone protein HSP90 in monocytes that are treated by MDP (Fig. 2d). In contrast, pretreating THP1-cells with MG132 strongly inhibited the formation of the NOD2/HSP90 complex in response to MDP, while LPS stimulation expectedly failed to do so (Fig. 2e). Consequently, NOD2 was migrating as a smear when treating THP-1 cells with MDP (Supplementary Figure 3a), in which a greater amount of NLRP12 was detected (Supplementary Figure 3b). Collectively, we identified NLRP12 as a NOD2-interacting protein that may promote NOD2 degradation and subsequently tolerance of infiltrating monocytes towards MDP."

3) In figure 3, the authors elegantly screen disease variants of NLRP12 and show that the NLRP12 R284X mutant fails to suppress NOD2-dependent NF- κ B activation, and human PBMCs with the NLRP12 mutation have enhanced IL-1 β production in response to MDP. While these studies provide important clinical relevance to the authors main conclusion that NLRP12 acts as a negative regulator of NOD2, there is some confusion as to how the R284X mutant fails to prevent NOD2 activity, while still retaining the ability to associate with NOD2 (Fig. 3d). The authors should reconcile this by determining if the NLRP12 mutant fails to induce NOD2 ubiquitination, similar to the experiment presented in Figure 2a. In addition, the protein levels of the R284X mutant appears to be undetectable in the western blots shown in Fig. 3c-d, which suggest the possibility that this mutant fails to suppress NOD2 because it is no longer expressed. This would conflict with the

authors results presented in Fig. 3d and conclusion that this mutant associates with NOD2. Alternatively, is the R284X mutant reduced in size, as suggested by the immunoblot where the authors immunoprecipitated and then immunoblotted NLRP12 (Fig. 3d, lower immunoblot, 4th lane)? Can the protein levels of the R284X mutant only be detected when first immunoprecipitated (opposed to immunoblotting the inputs)? Regardless, the authors should address this concern more thoroughly.

Author response. Additional transfection studies in HEK293 cells led us to confirm that the R284X mutant is reduced in size (new Fig. 3a) and fails to interact with the chaperone HSP90 (new Fig. 3d) that is required for NOD2 activity in myeloid cells (J Biol Chem. 2012 Nov 16; 287(47): 39800–39811). These results indicate that NLRP12 may repress NOD2 activity by transiently sequestering HSP90 in response to MDP. The statement in the text has been edited for better clarity and now reads : "All mutants with single amino-acid replacements were found to efficiently repress MDP-induced activation of NF- κ B (Fig. 3b) and to interact with HSP90 (Fig. 3c), as what observed in cells expressing wild-type NLRP12. In contrast, the R284X nonsense mutation fails to inhibit NOD2 signaling in response to MDP (Fig. 3b) and to recruit HSP90 (Fig. 3d), providing a potential explanation for the loss of tolerance to MDP in such mutant cells."

*4) In figure 5, the authors generate bone marrow chimeras to determine the cellular compartment expressing NLRP12 that is responsible for regulating ISGs in the intestine and provide support for a myeloid expression of NLRP12 in regulating this. Although this is consistent with previous reports describing the role of NLRP12 in inflammatory bowel disease and colitis-associated colon cancer, the authors should provide further functional assessments of myeloid vs. non-myeloid NLRP12 in vivo by assessing *C. rodentium* colonization in bone marrow chimeric mice, which would better support the overall conclusion of the manuscript, which focuses on bacterial tolerance and colonization.*

Author response. We thank the reviewer for underscoring that the findings are of interest. In the revised manuscript, we made all possible efforts and generated several sets of bone marrow chimeric mice that were subsequently infected by *C. rodentium*. While engraftment of lethally irradiated mice with the bone marrow from *Nlrp12*-deficient mice improved intraluminal elimination of *C. rodentium*, this protective effect was blunted in chimeric mice that were reconstituted with hematopoietic cells from mice that are deficient for both *Nod2* and *Nlrp12*. In contrast, the non-hematopoietic compartment of *Nlrp12*-deficient mice expectedly failed to contribute to the greater colonization resistance towards *C. rodentium*. These novel *in vivo* findings fully support our initial

conclusions. The results thereof are shown in the revised panel b of the seventh figure, and are discussed in the revised manuscript as follow: "In line with our previous findings, intraluminal elimination of *C. rodentium* was improved in lethally irradiated mice that were specifically engrafted with the bone marrow of *Nlrp12*-deficient mice although this difference was not statistically significant (Fig. 7b). Conversely, the colonization resistance was found similar in control and mutant mice that were reconstituted with wild-type bone marrow cells (Fig. 7b)".

5) *The authors demonstrate that $Nlrp12^{-/-}/Nod2^{-/-}$ double deficient mice have increased bacterial burdens compared to wild type mice (and $Nlrp12^{-/-}$ single deficient mice based on Fig. 6c). It would be experimentally challenging to measure NOD2 ubiquitination in *Nlrp12*-sufficient and deficient mice during *C. rodentium* infection; however it would be important to determine if NLRP12 regulates NOD2 protein expression in vivo to fully support the claim that NLRP12 promotes NOD2 degradation to regulate bacterial colonization. In addition, the authors need to measure NF- κ B activation in the colons of *C. rodentium*-infected mice as a measure of down-stream NOD2 activation.*

Author response. Although we agree that measuring NOD2 levels and NF- κ B activation *in situ* would further support our data, this is indeed a technically very challenging task with the commercially available antibodies and the limited material from mice. To overcome this issue, we generated a monocytic THP-1 cell line stably expressing the fusion protein Myc-BirA*-NOD2. This led us to confirm an endogenous interaction of NLRP12 with NOD2 in monocytes when specifically inhibiting the proteasome degradation of NOD2 that is induced in response to MDP (Fig. 1a). The additional data on THP-1 cells have been added in figure 2d-e and supplementary figure 3. In addition, our additional western-blotting analysis of NF- κ B activation led us to fully support our claim that NLRP12 function as a physiological checkpoint inhibitor of NOD2 signaling in monocytes. The dedicated paragraph has now been revised as follow: "Meanwhile, a greater MDP-induced NF- κ B activation was observed in monocytes from the bone marrow of *Nlrp12*-deficient mice but not in those from *Nlrp12^{-/-}:Nod2^{-/-}* mice (Supplementary Figure 14). Likewise, similar findings were observed in macrophages that were derived from the bone marrow of *Nlrp12*-deficient mice (Supplementary Figure 15), even if the difference was less pronounced when compared to what observed in monocytes (Supplementary Figure 14). A potential explanation for this difference may result from the down-regulation of NLRP12 during differentiation of monocytes into macrophages (data not shown and ²²)."

Minor Points

6) *In figure 1b, RIPK2 and NOD2 expression are extremely low or undetectable in the “input” immunoblots, despite both of these proteins being overexpressed and easily visible in the “IP” immunoblots. A darker exposure needs to be included to confirm expression and to accurately compare the IP blots.*

Author response. The quality of blots from the panel c of figure 1 has now been improved.

7) *In the results section, the figures 3c+3d are switched.*

Author response. The panels of figure 3 have now been correctly labeled.

8) *Please add labels to the bar graphs shown in Figure 5b to clarify which experimental group these graphs are referring to.*

Author response. The bar graphs shown in figure 5 have now been correctly labeled with each experimental groups.

9) *On page 10, lines 287-288, the authors state “we next assessed whether the transcriptional response of the intestine of mutant mice may potentially confer a growth advantage to *C. rodentium*” despite showing that *C. rodentium* is cleared more efficiently in *Nlrp12*-deficient mice in the previous figure (Fig. 6c). It seems the hypothesis should be framed in the opposite direction to fit with the presented data.*

Author response. Indeed, this has been an excellent suggestion. The result section has been revised as follow: “To further examine the basis of how the *Nlrp12*-mediated signaling pathway may compromise host defense against enteric bacterial pathogen infection, we next assessed whether the transcriptional response of the intestine of mutant mice may potentially confer colonization resistance to *C. rodentium*.”

10) *The authors describe colon crypt lengthening (hyperplasia) as an indicator of disease in Fig. 5. Both hyperplasia and immune infiltrate can be semiquantitatively scored in IHC-stained colons by a trained pathologist. This would allow the authors to determine if significant differences are present for these disease parameters. Similarly in Figure 6e and supplementary figure 8, a shorten colon and amplified *ERK1/2* pathways were shown which usually indicate an increased intestinal inflammation. Do the authors have data on weight loss observation and histopathological score for the intestine to indicate colitis?*

Author response. Even if histopathological scores for the intestine were already included in the previous version of the manuscript, we agree with the reviewer that data on weight loss observation would add to the strength of the findings. However, this parameter was not systematically monitored.

11) *In figure 7g, do the authors have growth curves for C. rodentium-infected Nlrp12^{-/-} mice to better directly compare these mice to the infected wild type and Nlrp12^{-/-}:Nod2^{-/-} double deficient animals, similar to the data presented in Fig. 7h-j.*

Author response. We also agree with this referee on this point. The bacterial load of *C. rodentium* in *Nlrp12^{-/-}* mice is now depicted in the new panel c from the last figure.

Reviewer #2: *In the current manuscript, authors demonstrated a role of NLRP12 in NOD2 signaling and bacterial tolerance using cells from FCAS2 patients and NLRP12-KO mice. This study is interesting in terms of the extension of role of NLRP12. However, there are several points are required to be addressed by authors.*

1. *Overall flow of description in this study. In the beginning, authors showed data from transfection study showing NLRP12-NOD2 connection. Actually authors showed suppression of NOD2 degradation and NF-κB activation. Then authors demonstrated FCAS2 patients and various mouse models. In FCAS2 patients and NLRP12-KO mice, authors focused on ISGs without showing the degradation of NOD2 and NF-κB activation, which should be provided for convincing story.* **Author response.** Although we agree that measuring NOD2 levels and NF-κB activation *in situ* would further support our data, this is indeed a technically very challenging task with the commercially available antibodies and the limited material from mice. To overcome this issue, we generated several clones of THP1 cells in which the expression of NLRP12 was specifically ablated by CRISPR/Cas9 system. This led us to confirm that "loss of NLRP12 expression by CRISPR/Cas9 system in human monocytic THP-1 cells enhanced secretion of TNF-alpha in response to MDP when compared to parental cells (Fig. 3g)". In addition, our additional western-blotting analysis of NF-κB activation led us to fully support our claim that NLRP12 function as a physiological checkpoint inhibitor of NOD2 signaling in monocytes. The dedicated paragraph has now been revised as follow: "Meanwhile, a greater MDP-induced NF-κB activation was observed in monocytes from the bone marrow of *Nlrp12*-deficient mice but not in those from *Nlrp12^{-/-}:Nod2^{-/-}* mice (Supplementary Figure 14). Likewise, similar findings were observed in macrophages that

were derived from the bone marrow of *Nlrp12*-deficient mice (Supplementary Figure 15), even if the difference was less pronounced when compared to what observed in monocytes (Supplementary Figure 14). A potential explanation for this difference may result from the down-regulation of NLRP12 during differentiation of monocytes into macrophages (data not shown and ²²). In addition, we Lastly, the text has also been edited for better clarity by homogeneizing the readouts on cytokines production, NF- κ B activation, STAT1 phosphorylation and the several IFN-stimulated genes that have been used in different experimental settings.

2. Figure 1. Demonstrated interaction of NLRP12 with NOD2 through linker region of NLRP12. But the construct without PYD-Linker also interacts (Fig 1c), which suggested involvement of multiple domains for the interaction.

(a) Thus it is needed to show which remaining domains (NBD or LRR) are involved in the NLRP12-NOD2 interaction.

Author response. We apologize if the cloning strategy was not clear. The NLRP12 dPYD Linker start with amino acid 200 of the NLRP12 protein (Proteomic databases P59046), which correlates to base pair 601 of its Open Reading Frame. The scheme has now been clarified and the text has been revised by mentioning that the construct without PYD-Linker shares 24 residues with the construct referred as 1-224. These residues form the ATP/GTP-specific phosphate binding loop called Walker A and a Mg²⁺ coordination site called Walker B. This agrees with previous reports indicating a suppressive function of ATP-binding domain on the activity of the plant R protein I-2 that shows similarities with NLRP1242 and may contribute to conformational changes of HSP90.

(b) Author should provide endogenous NLRP12 and NOD2 interaction.

Author response. We thank this reviewer for pointing this out. Given the poor sensitivity of commercially available NOD2 antibodies, we generated a monocytic THP-1 cell line stably expressing the fusion protein Myc-BirA*-NOD2 by using a retroviral system. This led us to confirm an endogenous interaction of NLRP12 with NOD2 in human monocytic cells when specifically inhibiting the MDP-induced proteasomal degradation of NOD2. These additional data are shown in the novel panel a of the first Figure. In line with this finding, the relevance of this interaction is illustrated by a set of additional *in vivo* experiments, that revealed a delayed clearance of *Citrobacter rodentium* in mice with specific depletion of Nod2 signaling in myeloid cells (new figure 7a). Conversely, a greater elimination of *C. rodentium* was observed in lethally irradiated mice that

were specifically engrafted with the bone marrow of *Nlrp12*-deficient mice. The text has been revised accordingly.

(c) The quality of Fig.1c is poor: in input blot, the bands are very faint and the amount of NLRP12 is not increased accordingly.

Author response. A longer exposure of these immunoblots has now been included. It is clear that the amount of NLRP12 is increasing within the ip NOD2 fraction, while being relatively stable in the input. This reason for this result remains unclear but has been observed several times. However, a reduced ubiquitinylation of RIPK2 and a lowered total amount of NOD2 were clearly observed when transfecting more NLRP12. This finding further support our main conclusion that the NOD2/RIPK2 complex is prone for proteosomal degradation in cells overexpressing NLRP12.

3. Figure 2c, lack of controls: should include NF- κ B activation in the presence of various NLRP12 without NOD2.

Author response. We apologize for having omitted to include this important control. In line with previous report showing that NLRP12 overexpression failed to induce NF- κ B activation (Williams et al, JBC 2005), the panel c of the second Figure now depicts that NF- κ B activation is not observed when transfecting NLRP12 alone even at its highest concentration.

4. For Figure 3

(a) Figure 3a, if the FCAS2 patients are the same patients reported in the previous publication (ref #1), this panel should be removed from main figures and just mention in text with reference.

Author response. The pedigree of the family has now been deleted from the revised figure 3 and the previous publication has now been referred to accordingly.

(b) In FCAS patients with R284X, authors insisted haploinsufficiency rather than dominant-negative effect of the mutant NLRP12 (Fig 3f). To prove that, author should show the level of NLRP12 at the endogenous protein level by Western blot in comparing with healthy controls.

Author response. In line with our results showing that nonsense-mediated mRNA decay was activated in cells from FCAS patients with R284X (Fig. 3f), we failed to observe the mutant NLRP12 protein in peripheral blood monocytes from FCAS patients with R284X (data available upon request by the reviewer). Foremost, ectopic expression of NLRP12 R284X in HEK293T cells clearly showed that this polypeptide produced by this gene was rather unstable compared to full-

length NLRP12 even when transfecting twice more quantity of the mutant plasmid (Fig. 3a). Lastly, we performed additional transfecting studies in HEK293 cells and the dedicated paragraph has been revised as follow: "In line with the mapping of the interaction between NOD2 and NLRP12, the R284X nonsense mutation was found to interact with NOD2 (Fig. 3a). All mutants with single amino-acid replacements were found to efficiently repress MDP-induced activation of NF- κ B (Fig. 3b) and to interact with HSP90 (Fig. 3c), as what observed in cells expressing wild-type NLRP12. In contrast, the R284X nonsense mutation fails to inhibit NOD2 signaling in response to MDP (Fig. 3b) and to recruit HSP90 (Fig. 3d), providing a potential explanation for the loss of tolerance to MDP in such mutant cells. Consequently, MDP induced a greater secretion of interleukin-6 (IL-6) and interleukin-1beta (IL1b) by the peripheral blood mononuclear cells from patients bearing the R284X nonsense mutation when compared to control cells (Fig 3e and data not shown)". Collectively, these results further argue against a dominant negative effect of this mutant protein and we now provide a sentence on the key role of nonsense-mediated mRNA decay in eliminating mRNAs containing premature translation-termination codons for clarity.

(c) It is not clear what is the role of Hsp90 on NLRP12 and NF-kB activation. In the previous data, author demonstrated that NLRP12 suppresses NOD2-mediated NF-kB activation through NLRP12-NOD2 interaction and degradation. However, the R284X mutant NLRP12 is still binding with NOD2 without NF-kB suppression. In contrast, R352C mutant shows decreased binding with NOD2 and suppressing NF-kB activation as much as WT NLRP12. This result is not consistence with previous results. In addition, author should show the level of NOD2 in patients and healthy controls by Western blot whether it is reduced or not because the R284X-NLRP12 mutant probably fails for degrading NOD2.

Author response. To assess whether R352C mutant shows decreased binding with NOD2, we have performed a large series of reverse co-immunoprecipitation experiments. In line with our previous finding on the suppressive activity of R352C mutant on MDP-induced NF- κ B activation, the new panel a and c of figure 3 now depicts a similar interaction of R352C mutant with NOD2 and HSP90 respectively. Furthermore, co-immunoprecipitation experiments confirmed that NOD2 endogenously interacts with the chaperone protein HSP90 in monocytes that are treated by MDP (Fig. 2d). In contrast, pretreating THP1-cells with MG132 strongly inhibited the formation of the NOD2/HSP90 complex in response to MDP, while LPS stimulation expectedly failed to do so (Fig. 2e). Consequently, NOD2 was migrating as a smear when treating THP-1 cells with MDP (Supplementary Figure 3a), in which a greater amount of NLRP12 was detected (Supplementary

Figure 3b). For clarity, the introduction has been revised as follow: "Herein, we report evidence that NLRP12 promotes bacterial tolerance by repressing NOD2 signaling, while the FCAS2-causing NLRP12 mutation fails to sequester the heat-shock protein 90 (HSP90) that is one of the most abundant molecular chaperone in the cytosol. Loss of Hsp90 complex results in MDP tolerance through a failure to protect NOD2 from being degraded by the proteasome¹⁷. This loss of tolerance is exemplified by a greater recruitment of inflammatory monocytes that protects the epithelium of Nlrp12-deficient mice from attaching-and-effacing bacterial pathogen through induction of several interferon-stimulated genes."

(d) Fig.3c is not appropriate. R284X is smaller than WT or other missense mutant NLRP12. Thus author should show whole range of blot in this figure.

Author response. We repeated this experiment and the lower part of the gel is now shown in the revised figure 3d (formerly referred as figure 3c).

5. Figure 4.

(a) High level of LPS challenge results in similar susceptibility but low LPS and MDP challenging shows significant difference. In this experiment, low dose LPS challenging without MDP is missing. This is an important control.

Author response. We agree with the reviewer that this is an important control to be included. The survival curve of mice that received only low dose of LPS has now been added in the revised figure 4b (blue survival curve).

(b) Regarding to FCAS2 patients with R284X, which is loss-of-function and haploinsufficiency, author should include heterozygotes (Nlrp12+/-) in this challenging experiments.

Author response. We are sorry that our efforts in attempting to demonstrate haploinsufficiency was not clear enough for this referee. We agree that performing additional experiments using heterozygous mice would further strength this point on haploinsufficiency. However, our European animal welfare regulation would not allow us to perform such experiment at this stage.

(c) In this figure, only Cxcl1 levels are provides (Fig. 4a), while FCAS2 patients with R284X has shown increased IL-1beta. Same readings are needed in two related systems (patients and animal model) to make consistent story.

Author response. The dataset has now been homogenized by including IL-6 level in both Figure 3e and 4d. Furthermore, we have also quantified the level of TNF-alpha in the supernatant of peripheral blood mononuclear cells from patients and controls by specific ELISA (data available upon request) and of the newly generated THP1 clones (Fig. 3g), but it was no detectable in the serum of septic mice.

6. Figure 5: *What's the NOD2 levels in NLRP12-KO mice? If all of the ISG increases are due to NOD2 activation in NLRP12-KO mice, that should be proved. It is very important in context of whole story of this study.*

Author response. We have to state clearly that the level of Nod2 is not robustly detected in the colon of mice due to technical reasons. However, the dependence of the enhanced ISG response

on Nod2 level is demonstrated by making use of mice that are deficient for both Nod2 and Nlrp12. In line with this finding shown in the last panel of figure 7, we have included additional qRTPCR data showing that STAT1 phosphorylation and the expression of Ifit2, Ifit44, Oas2 and Apol9ab were lowered in the colon of Nod2-deficient mice at steady state (new supplementary figure 13a-b).

7. *Figure 6: in addition to Fig. 6b (Ifit2-KO) and 6c (Nlrp12-KO), bacterial colonization of double KO for Ifit2 and Nlrp12 is needed to confirm directly that reduced colonization of Nlrp12-KO mice is due to increased Ifit2 expression.*

Author response. The reviewer raises an important point, yet we have to clarify that Ifit2 is likely not the only downstream factor that protects *Nlrp12*-deficient mice toward *C. rodentium*. This is why we monitored the growth of *C. rodentium* in mice that are deficient for both Nlrp12 and Nod2 in which the level of Ifit2 was found similar to wild-type mice (new Figure 7f). The growth curves for *C. rodentium*-infected *Nlrp12*^{-/-} mice have also been depicted in the new panel c from the last figure. In this context, we consider that it may take more than a year to generate additional knockout mice for all relevant ISG (and not only Ifit2).

8. *In supplementary figure 4, author showed hyper-activation of STAT1, which leads ISG induction. However, in supplementary figure 12, defect of STAT3 is shown. If both STAT1 and 3 are regulated by NOD2 activation. Which signal is important, STAT1 or STAT3, or both? Author should show the status of STAT1 and 3 in three KO mice, NLRP12-, Ifit2-, and NOD2-KO mice.*

Author response. We agree with the referee that we cannot conclude that either STAT1 or STAT3 activation would contribute to the protection of *Nlrp12*-deficient mice. However, we must admit that STAT3 activation is not likely contributing to the greater ISG induction in *Nlrp12*-deficient mice, as we failed to observe changes in phosphorylation of STAT3 in macrophages from *Nlrp12*-deficient mice that were treated by MDP (data available upon request). Therefore, the panel a of the new supplementary figure 13 has been replaced by additional western blotting analysis of STAT1 phosphorylation in the intestine of wild-type and *Nod2*-deficient mice and the text has been revised accordingly to avoid any confusion for the readers.

Minor points:

1. *The labeling of Figure 3b and 3c is not match in text (page 6, lines 137 and 139).*

Author response. The numbering of all figures has now been verified carefully in the revised version of the manuscript.

2. Legend for Figure 4c is missing.

Author response. We apologize for this omission. The legend of Figure 4c has now been added in the revised version of our manuscript.

3. Supplementary table 1 is missing.

Author response. We apologize for this omission. The supplementary table 1 has now been submitted together with revised version of our manuscript.

Reviewers' comments:

Reviewer #1 (Remarks to the Author):

In the revised version of the manuscript "Proteosomal degradation of Nod2 by Nlrp12 promotes bacterial tolerance and colonization by pathogen" the authors include new experiments that provide greater mechanistic clarity and physiological relevance to their presented findings describing how NLRP12 regulates NOD2 expression and function during intestinal colonization. This work is important in that it contributes to our understanding of how immune sensors function as negative regulators of innate immune responses. Specifically, the authors provide new data showing:

- 1) NLRP12 interacts with NOD2 in human monocytic cells (THP-1) stimulated with the NOD2 ligand MDP, and both NLRP12 and NOD2 expression in myeloid cells regulates *C. rodentium* clearance in vivo;
- 2) NOD2 ubiquitination is observed following MDP stimulation in the presence of MG-132, and MG-132 prevents NLRP12-mediated loss of NOD2 protein;
- 3) The disease-associated R284X mutant of NLRP12 displays a reduced ability to interact with HSP90, which has previously been shown to be required for NOD2 stability; thus, these findings provide a mechanistic link between NLRP12, NOD2 stability and loss of tolerance to MDP stimulation with the R284X disease mutant;
- 4) Reduced NF- κ B activation in *Nlrp12^{-/-}/Nod2^{-/-}* double-deficient monocytes compared to *Nlrp12^{-/-}* monocytes following MDP stimulation.

The new co-immunoprecipitation studies presented in Figure 1a and Figure 2e provide important mechanistic insight into the regulation of NOD2 by NLRP12, these results could benefit from proper input controls in order to make the claim that the changes in the described interacting proteins are not solely due to the overall loss/reduction of the target proteins. However this should not deter acceptance of the work.

Reviewer #2 (Remarks to the Author):

The manuscript, entitled "Proteosomal degradation of Nod2 by Nlrp12 promotes bacterial tolerance and colonization by pathogen" describes a role of NLRP12 in NOD2 signaling and bacterial tolerance. NLRP12 induces proteosomal degradation of NOD2 that mediates MDP-induced inflammation. Thus, deficiencies of NLRP12 (by KO or R284X mutation) result in defect suppression of NOD2, which leads intestinal or colon inflammation. Authors also demonstrated that the inflammation is mediated by ISG induction. This is the revised version and the authors answered to the comments raised from previous version of manuscript. Nevertheless, there are still few critic points that should be addressed to recommend an acceptance for publication. Also, there are several additional questions to be fixed or addressed.

1. Inconsistencies of cytokine productions. The patients' PBMCs with R284X showed increased IL-6 and IL-1 β in response to MDP (Fig.3e and author did not provide for IL-1 β). In the other hand, authors provide only TNF- α from NLRP12-KO THP-1 cells (Fig3g) and mentioned that MDP-induced IL-6 was barely detectable in the text. Then, from the studies of NLRP12-KO mice, authors focused on the genes regulated by IFN (Type I/III), and found that the increased expression of ISG are induced by NLRP12-KO leukocytes, which is proved by bone marrow transfer experiments (Fig. 5). From the provided data in the manuscript, it is ambiguous which cytokine is regulated by NOD2-signaling without NLRP12 suppression. What kind of cytokines are abnormally secreted from the NLRP12-KO leukocytes in response to MDP? Since NOD2 and NLRP12 are reported to be involved an inflammasome activation, the IL-1 β level is important. Thus, it is very important to show same cytokine profiles from NLRP12-KO THP1 cells, patients' PBMCs with R284X, and NLRP12-KO leukocytes, and to find any connection for ISG expression.
2. The increased susceptibility of NLRP12-KO mice to MDP (Fig. 4a) and the increased expression of IFIT2 in colon of NLRP12-KO mice (Fig. 5d) are probably due to NOD2 that is not suppressed by

NLRP12. However, the data provided in Fig. 4 and 5 are not sufficient for the role of NOD2. Thus, in Fig.4 and 5, author should include the data from NLRP12^{-/-} NOD2^{-/-} (double KO) mice that were used Fig.7.

3. Also, it is necessary to show the bacterial colonization data from NLRP12^{-/-} Ifit2^{-/-} mice in addition to individual NLRP12^{-/-} or Ifit2^{-/-} mice (Fig.6b and c), which will be the direct evidence the deficiency of NLRP12 results in less colonization through Ifit2.

4. Immunoblot for the NLRP12 from patients' monocytes should be provided. This data is not only for the mutant band that probably smaller than WT band or not detectible, but also for reduced level of WT NLRP12, which also necessary to support haploinsufficiency.

5.To show endogenous interaction of NOD2 and NLRP12, authors generated stable cell THP-1 cell line expressing Myc-NOD2 (Fig. 1a). This data supports the NOD2-NLRP12 interaction in monocytic cell line, but it does not support an endogenous interaction because Myc-NOD2 is not endogenous protein. In addition, authors showed inappropriate input, 'tubulin' in co-immunoprecipitation data of Fig. 1a and Fig. 2e. For Fig. 1a, input should be Myc-NOD2 and NLRP12, and for Fig. 2e, Myc-NOD2 and HSP90.

6. The immunoprecipitation data for R284X NLRP12 with Hsp90 (Fig. 3d) is not reliable. In the input data, the level of R284X NLRP12 is much less (even non-detectible) than WT NLRP12. The defect of R284X-Hsp90 is probably due to the lower level of R284X.

7. The data shown in Fig. 4c and d are not reliable. Statistic differences were observed at only one-time point (1.5h), and even opposite in different time points, which are not significant (probably due to small number of animals used in these experiments).

8. For the comment #6- in Figure 5, what's the NOD2 levels in NLRP12-KO mice? If all of the ISG increases are due to NOD2 activation in NLRP12-KO mice, that should be proved. It is very important in context of whole story of this study.

Author response. We have to state clearly that the level of Nod2 is not robustly detected in the colon of mice due to technical reasons.

Additional comment: NOD2 is mainly expressed in leukocytes but not in colon. The previous comment was requesting to show level of NOD2 in monocytes or BMDMs of NLRP12^{-/-} and NLRP12^{+/+}.

9. Supplementary Fig. 14, total IκBα also should be shown with pIκBα.

10. Fig. 5g is not mentioned in the text.

Point-by-point response to reviewers' comments

Reviewer #1: Reviewer #1 (Remarks to the Author):

In the revised version of the manuscript "Proteosomal degradation of Nod2 by Nlrp12 promotes bacterial tolerance and colonization by pathogen" the authors include new experiments that provide greater mechanistic clarity and physiological relevance to their presented findings describing how NLRP12 regulates NOD2 expression and function during intestinal colonization. This work is important in that it contributes to our understanding of how immune sensors function as negative regulators of innate immune responses. Specifically, the authors provide new data showing:

- 1) NLRP12 interacts with NOD2 in human monocytic cells (THP-1) stimulated with the NOD2 ligand MDP, and both NLRP12 and NOD2 expression in myeloid cells regulates C. rodentium clearance in vivo;*
- 2) NOD2 ubiquitination is observed following MDP stimulation in the presence of MG-132, and MG-132 prevents NLRP12-mediated loss of NOD2 protein;*
- 3) The disease-associated R284X mutant of NLRP12 displays a reduced ability to interact with HSP90, which has previously been shown to be required for NOD2 stability; thus, these findings provide a mechanistic link between NLRP12, NOD2 stability and loss of tolerance to MDP stimulation with the R294X disease mutant;*
- 4) Reduced NF- κ B activation in Nlrp12^{-/-}/Nod2^{-/-} double-deficient monocytes compared to Nlrp12^{-/-} monocytes following MDP stimulation. The new co-immunoprecipitation studies presented in Figure 1a and Figure 2e provide important mechanistic insight into the regulation of NOD2 by NLRP12, these results could benefit from proper input controls in order to make the claim that the changes in the described interacting proteins are not solely due to the overall loss/reduction of the target proteins. However this should not deter acceptance of the work.*

Author response. We thank the first reviewer for her/his encouraging comments. Proper input controls for co-precipitation experiments have now been shown in order to make the claim that the changes in the described interacting proteins are not solely due to the overall loss/reduction of the target proteins.

Reviewer #2: (Remarks to the Author):

The manuscript, entitled "Proteosomal degradation of Nod2 by Nlrp12 promotes bacterial tolerance and colonization by pathogen" describes a role of NLRP12 in NOD2 signaling and bacterial

tolerance. NLRP12 induces proteosomal degradation of NOD2 that mediates MDP-induced inflammation. Thus, deficiencies of NLRP12 (by KO or R284X mutation) result in defect suppression of NOD2, which leads intestinal or colon inflammation. Authors also demonstrated that the inflammation is mediated by ISG induction. This is the revised version and the authors answered to the comments raised from previous version of manuscript.

Author response. We thank the reviewer for her/his appraisal.

Nevertheless, there are still few critic points that should be addressed to recommend an acceptance for publication. Also, there are several additional questions to be fixed or addressed

1. Inconsistencies of cytokine productions. The patients' PBMCs with R284X showed increased IL-6 and IL-1beta in response to MDP (Fig.3e and author did not provide for IL-1beta). In the other hand, authors provide only TNF-alpha from NLRP12-KO THP-1 cells (Fig3g) and mentioned that MDP-induced IL-6 was barely detectable in the text. Then, from the studies of NLRP12-KO mice, authors focused on the genes regulated by IFN (Type I/III), and found that the increased expression of ISG are induced by NLRP12-KO leukocytes, which is proved by bone marrow transfer experiments (Fig. 5). From the provided data in the manuscript, it is ambiguous which cytokine is regulated by NOD2-signaling without NLRP12 suppression. What kind of cytokines are abnormally secreted from the NLRP12-KO leukocytes in response to MDP? Since NOD2 and NLRP12 are reported to be involved an inflammasome activation, the IL-1beta level is important. Thus, it is very important to show same cytokine profiles from NLRP12-KO THP1 cells, patients' PBMCs with R284X, and NLRP12-KO leukocytes, and to find any connection for ISG expression.

Author response. We thank the reviewer for pointing to the necessity of improving the overall flow of description in this study to the broader readership of *Nature Communication* as it may seem counter-intuitive. The introduction section has first been improved with greater details on the experimental evidences showing that NLRP12 negatively regulates the activation of either canonical or non-canonical NF- κ B pathway. While being consistent with our microarray and qRTPCR data in mice, the use of a type I/III interferon-specific reporter cell line-based bioassay led us to confirm *in vitro* that NLRP12 negatively regulates type I/III IFN signaling pathway in human cells, while the NLRP12 R284X mutation failed to repress induction of such anti-viral response. We next performed a more comprehensive evaluation on such novel regulatory role of NLRP12 by focusing on tumor necrosis factor alpha (TNF-alpha) and interleukin-6 (IL-6) that are known to regulate Type I/III IFN signaling pathway (Immunity. 2006 Sep;25(3):349-60). In agreement with our NF- κ B reporter assay (new Figure 4a) and with data from NLRP12-deficient THP1 cells (new

Figure 4e), the secretion of both TNF-alpha and IL-6 by monocytes from mutant mice is increased in response to MDP as a closer model to the THP-1 cell line (Supplementary Figure 7b). The levels of both TNF-alpha and IL-6 in experimental settings of patients' PBMCs is now reported in the revised Figure 4f and Supplementary Figure 6b respectively. Finally, the molecular interplay between the aforementioned pathway and type I/III interferon signaling pathway is now explained in a more didactic way and with more details o, the revised version of the manuscript.

2. The increased susceptibility of NLRP12-KO mice to MDP (Fig. 4a) and the increased expression of IFIT2 in colon of NLRP12-KO mice (Fig. 5d) are probably due to NOD2 that is not suppressed by NLRP12. However, the data provided in Fig. 4 and 5 are not sufficient for the role of NOD2. Thus, in Fig.4 and 5, author should include the data from *Nlrp12^{-/-} Nod2^{-/-}* (double KO) mice that were used Fig.7.

Author response. Additional data have now been carefully included in the new figure 3 that now depicts a survival curve demonstrating that the increased susceptibility of *Nlrp12*-deficient mice to MDP strictly relies on NOD2 expression (new Figure 3c). Encouraged by these results, the figure 4 has also been improved by including the qRT-PCR data showing a NOD2-dependent expression of ISG in the intestine of *Nlrp12*-deficient mice (new Figure 4e).

3. Also, it is necessary to show the bacterial colonization data from NLRP12^{-/-} Ifit2^{-/-} mice in addition to individual NLRP12^{-/-} or Ifit2^{-/-} mice (Fig.6b and c), which will be the direct evidence the deficiency of NLRP12 results in less colonization through Ifit2.

Author response. This is an interesting question, which remains beyond the scope of the present study. Indeed, we did not aim to overstate the effect of IFIT2, but rather to suggest that the greater secretion of IFIT2 in the absence of NLRP12 may contribute (either alone or in conjunction with other antimicrobial peptides) to the greater colonization resistance as a consequence of activation of NOD2 signaling. For clarity, the title of the result section describing the data using Ifit2-deficient mice now reads 'NLRP12 deficiency contributes to a greater colonization resistance against attaching-and-effacing bacterial pathogen through activation of NOD2 signaling in monocytes'.

4. Immunoblot for the NLRP12 from patients' monocytes should be provided. This data is not only for the mutant band that probably smaller than WT band or not detectible, but also for reduced level of WT NLRP12, which also necessary to support haploinsufficiency.

Author response. Immunoblot for NLRP12 from patients' PBMCs is now provided (new Supplementary Figure 6). This confirms that the R284X mutation results in haploinsufficiency through activation of mRNA decay.

5. To show endogenous interaction of NOD2 and NLRP12, authors generated stable cell THP-1 cell line expressing Myc-NOD2 (Fig. 1a). This data supports the NOD2-NLRP12 interaction in monocytic cell line, but it does not support an endogenous interaction because Myc-NOD2 is not endogenous protein. In addition, authors showed inappropriate input, 'tubulin' in co-immunoprecipitation data of Fig. 1a and Fig. 2e. For Fig. 1a, input should be Myc-NOD2 and NLRP12, and for Fig. 2e, Myc-NOD2 and HSP90.

Author response. Proper input controls for co-precipitation experiments have now been shown in order to make the claim that the changes in the described interacting proteins are not solely due to the overall loss/reduction of the target proteins. Furthermore, the term 'endogenous' has been removed from the text according to the reviewer's comment.

6. The immunoprecipitation data for R284X NLRP12 with Hsp90 (Fig. 3d) is not reliable. In the input data, the level of R284X NLRP12 is much less (even non-detectable) than WT NLRP12. The defect of R284X-Hsp90 is probably due to the lower level of R284X.

Author response. We apologize if our interpretation was not clear enough in the previous version. We agree that the NLRP12 R284X mutant is expressed at very low levels, further supporting that its negative effect on NOD2 signaling is expected to be even stronger. Accordingly, we changed the text to better reflect this fact as follows: 'It coincided with a barely detectable recruitment of HSP90 by such mutation (Fig. 4c) even if this was not related to a failure of the R284X nonsense mutation to interact with NOD2 (Fig. 4d)'.

7. The data shown in Fig. 4c and d are not reliable. Statistical differences were observed at only one time point (1.5h), and even opposite in different time points, which are not significant (probably due to small number of animals used in these experiments).

Author response. According to the reviewer's comment, the data shown in the panel c and d of the previous figure 4 have now been removed from the revised version of this figure that is now labeled as figure 3.

8. For the comment #6- in Figure 5, what's the NOD2 levels in NLRP12-KO mice? If all of the ISG increases are due to NOD2 activation in NLRP12-KO mice, that should be proved. It is very important in context of whole story of this study.

Author response. Additional qRT-PCR analysis have now been included for demonstrating that all of the ISG increases are due to NOD2 activation in the intestine of *Nlrp12*-deficient mice. However, we have to state clearly that the level of NOD2 could not be robustly detected in the colon of mice in our hands due to technical reasons.

Additional comment: NOD2 is mainly expressed in leukocytes but not in colon. The previous comment was requesting to show level of NOD2 in monocytes or BMDMs of NLRP12^{-/-} and NLRP12^{+/+}.

9. Supplementary Fig. 14, total I κ B α also should be shown with pI κ B α .

Author response. We apologize for this omission but total I κ B α was barely detectable in monocytes that were isolated from the bone marrow of mice (new Supplementary Figure 7).

10. Fig. 5g is not mentioned in the text.

Author response. We apologize for this omission that has now been rectified.

REVIEWERS' COMMENTS:

Reviewer #1 (Remarks to the Author):

the authors have addressed all of the points from the prior review

Reviewer #2 (Remarks to the Author):

In the manuscript, entitled "Proteosomal degradation of NOD2 by NLRP12 in monocytes promotes bacterial tolerance and colonization by enteric pathogen", authors unveiled underlying mechanism for the role of NLRP12 in NOD2 signaling and bacterial tolerance using various animal models and a disease-causing NLRP12 mutation. This is the revised version and the authors answered to the comments raised from previous version of manuscript by providing additional data. This study is interesting and will provides a knowledge in understanding a part of immune response to pathogens.